# Online and Stochastic Gradient Methods for Non-decomposable Loss Functions

**Purushottam Kar**[*]     **Harikrishna Narasimhan**[†]     **Prateek Jain**[*]

[*]Microsoft Research, INDIA

[†]Indian Institute of Science, Bangalore, INDIA

{t-purkar,prajain}@microsoft.com, harikrishna@csa.iisc.ernet.in

## Abstract

Modern applications in sensitive domains such as biometrics and medicine frequently require the use of *non-decomposable* loss functions such as precision@$k$, F-measure etc. Compared to point loss functions such as hinge-loss, these offer much more fine grained control over prediction, but at the same time present novel challenges in terms of algorithm design and analysis. In this work we initiate a study of online learning techniques for such non-decomposable loss functions with an aim to enable incremental learning as well as design scalable solvers for batch problems. To this end, we propose an online learning framework for such loss functions. Our model enjoys several nice properties, chief amongst them being the existence of efficient online learning algorithms with sublinear regret and online to batch conversion bounds. Our model is a provable extension of existing online learning models for point loss functions. We instantiate two popular losses, Prec@$k$ and pAUC, in our model and prove sublinear regret bounds for both of them. Our proofs require a novel structural lemma over ranked lists which may be of independent interest. We then develop scalable stochastic gradient descent solvers for non-decomposable loss functions. We show that for a large family of loss functions satisfying a certain uniform convergence property (that includes Prec@$k$, pAUC, and F-measure), our methods provably converge to the empirical risk minimizer. Such uniform convergence results were not known for these losses and we establish these using novel proof techniques. We then use extensive experimentation on real life and benchmark datasets to establish that our method can be orders of magnitude faster than a recently proposed cutting plane method.

## 1   Introduction

Modern learning applications frequently require a level of fine-grained control over prediction performance that is not offered by traditional "per-point" performance measures such as hinge loss. Examples include datasets with mild to severe label imbalance such as spam classification wherein positive instances (spam emails) constitute a tiny fraction of the available data, and learning tasks such as those in medical diagnosis which make it imperative for learning algorithms to be sensitive to class imbalances. Other popular examples include ranking tasks where precision in the top ranked results is valued more than overall precision/recall characteristics. The performance measures of choice in these situations are those that evaluate algorithms over the entire dataset in a holistic manner. Consequently, these measures are frequently *non-decomposable* over data points. More specifically, for these measures, the loss on a set of points cannot be expressed as the sum of losses on individual data points (unlike hinge loss, for example). Popular examples of such measures include F-measure, Precision@$k$, (partial) area under the ROC curve etc.

Despite their success in these domains, non-decomposable loss functions are not nearly as well understood as their decomposable counterparts. The study of point loss functions has led to a deep

understanding about their behavior in batch and online settings and tight characterizations of their generalization abilities. The same cannot be said for most non-decomposable losses. For instance, in the popular online learning model, it is difficult to even instantiate a non-decomposable loss function as defining the per-step penalty itself becomes a challenge.

## 1.1 Our Contributions

Our first main contribution is a framework for online learning with non-decomposable loss functions. The main hurdle in this task is a proper definition of instantaneous penalties for non-decomposable losses. Instead of resorting to canonical definitions, we set up our framework in a principled way that fulfills the objectives of an online model. Our framework has a very desirable characteristic that allows it to recover existing online learning models when instantiated with point loss functions. Our framework also admits online-to-batch conversion bounds.

We then propose an efficient Follow-the-Regularized-Leader [1] algorithm within our framework. We show that for loss functions that satisfy a generic "stability" condition, our algorithm is able to offer vanishing $\mathcal{O}\left(\frac{1}{\sqrt{T}}\right)$ regret. Next, we instantiate within our framework, convex surrogates for two popular performances measures namely, Precision at $k$ ($\text{Prec}_{@k}$) and partial area under the ROC curve (pAUC) [2] and show, via a stability analysis, that we do indeed achieve sublinear regret bounds for these loss functions. Our stability proofs involve a structural lemma on sorted lists of inner products which proves Lipschitz continuity properties for measures on such lists (see Lemma 2) and might be useful for analyzing non-decomposable loss functions in general.

A key property of online learning methods is their applicability in designing solvers for offline/batch problems. With this goal in mind, we design a stochastic gradient-based solver for non-decomposable loss functions. Our methods apply to a wide family of loss functions (including $\text{Prec}_{@k}$, pAUC and F-measure) that were introduced in [3] and have been widely adopted [4, 5, 6] in the literature. We design several variants of our method and show that our methods provably converge to the empirical risk minimizer of the learning instance at hand. Our proofs involve uniform convergence-style results which were not known for the loss functions we study and require novel techniques, in particular the structural lemma mentioned above.

Finally, we conduct extensive experiments on real life and benchmark datasets with pAUC and $\text{Prec}_{@k}$ as performance measures. We compare our methods to state-of-the-art methods that are based on cutting plane techniques [7]. The results establish that our methods can be significantly faster, all the while offering comparable or higher accuracy values. For example, on a KDD 2008 challenge dataset, our method was able to achieve a pAUC value of 64.8% within 30ms whereas it took the cutting plane method more than 1.2 seconds to achieve a comparable performance.

## 1.2 Related Work

Non-decomposable loss functions such as $\text{Prec}_{@k}$, (partial) AUC, F-measure etc, owing to their demonstrated ability to give better performance in situations with label imbalance etc, have generated significant interest within the learning community. From their role in early works as indicators of performance on imbalanced datasets [8], their importance has risen to a point where they have become the learning objectives themselves. Due to their complexity, methods that try to indirectly optimize these measures are very common e.g. [9], [10] and [11] who study the F-measure. However, such methods frequently seek to learn a complex probabilistic model, a task arguably harder than the one at hand itself. On the other hand are algorithms that perform optimization directly via structured losses. Starting from the seminal work of [3], this method has received a lot of interest for measures such as the F-measure [3], average precision [4], pAUC [7] and various ranking losses [5, 6]. These formulations typically use cutting plane methods to design dual solvers.

We note that the learning and game theory communities are also interested in non-additive notions of regret and utility. In particular [12] provides a generic framework for online learning with non-additive notions of regret with a focus on showing regret bounds for mixed strategies in a variety of problems. However, even polynomial time implementation of their strategies is difficult in general. Our focus, on the other hand, is on developing efficient online algorithms that can be used to solve large scale batch problems. Moreover, it is not clear how (if at all) can the loss functions considered here (such as $\text{Prec}_{@k}$) be instantiated in their framework.

Recently, online learning for AUC maximization has received some attention [13, 14]. Although AUC is not a point loss function, it still decomposes over pairs of points in a dataset, a fact that [13] and [14] crucially use. The loss functions in this paper do not exhibit any such decomposability.

## 2 Problem Formulation

Let $\mathbf{x}_{1:t} := \{\mathbf{x}_1, \ldots, \mathbf{x}_t\}$, $\mathbf{x}_i \in \mathbb{R}^d$ and $y_{1:t} := \{y_1, \ldots, y_t\}$, $y_i \in \{-1, 1\}$ be the observed data points and *true* binary labels. We will use $\widehat{y}_{1:t} := \{\widehat{y}_1, \ldots, \widehat{y}_t\}$, $\widehat{y}_i \in \mathbb{R}$ to denote the predictions of a learning algorithm. We shall, for sake of simplicity, restrict ourselves to linear predictors $\widehat{y}_i = \mathbf{w}^\top \mathbf{x}_i$ for parameter vectors $\mathbf{w} \in \mathbb{R}^d$. A performance measure $\mathcal{P} : \{-1, 1\}^t \times \mathbb{R}^t \to \mathbb{R}_+$ shall be used to evaluate the the predictions of the learning algorithm against the true labels. Our focus shall be on non-decomposable performance measures such as $\text{Prec}_{@k}$, partial AUC etc.

Since these measures are typically non-convex, convex surrogate *loss functions* are used instead (we will use the terms *loss function* and *performance measure* interchangeably). A popular technique for constructing such loss functions is the *structural SVM* formulation [3] given below. For simplicity, we shall drop mention of the training points and use the notation $\ell_\mathcal{P}(\mathbf{w}) := \ell_\mathcal{P}(\mathbf{x}_{1:T}, y_{1:T}, \mathbf{w})$.

$$\ell_\mathcal{P}(\mathbf{w}) = \max_{\bar{\mathbf{y}} \in \{-1, +1\}^T} \sum_{i=1}^{T} (\bar{y}_i - y_i)\mathbf{x}_i^\top \mathbf{w} - \mathcal{P}(\bar{\mathbf{y}}, \mathbf{y}). \tag{1}$$

**Precision@k.** The $\text{Prec}_{@k}$ measure ranks the data points in order of the predicted scores $\widehat{y}_i$ and then returns the number of true positives in the top ranked positions. This is valuable in situations where there are very few positives. To formalize this, for any predictor $\mathbf{w}$ and set of points $\mathbf{x}_{1:t}$, define $S(\mathbf{x}, \mathbf{w}) := \{j : \mathbf{w}^\top \mathbf{x} > \mathbf{w}^\top \mathbf{x}_j\}$ to be the set of points which $\mathbf{w}$ ranks above $\mathbf{x}$. Then define

$$\mathbb{T}_{\beta,t}(\mathbf{x}, \mathbf{w}) = \begin{cases} 1, & \text{if } |S(\mathbf{x}, \mathbf{w})| < \lceil \beta t \rceil, \\ 0, & \text{otherwise.} \end{cases} \tag{2}$$

i.e. $\mathbb{T}_{\beta,t}(\mathbf{x}, \mathbf{w})$ is non-zero iff $\mathbf{x}$ is in the top-$\beta$ fraction of the set. Then we define[1]

$$\text{Prec}_{@k}(\mathbf{w}) := \sum_{j : \mathbb{T}_{k,t}(\mathbf{x}_j, \mathbf{w}) = 1} \mathbb{I}\left[y_j = 1\right].$$

The structural surrogate for this measure is then calculated as [2]

$$\ell_{\text{Prec}_{@k}}(\mathbf{w}) = \max_{\substack{\bar{\mathbf{y}} \in \{-1, +1\}^t \\ \sum_i (\bar{y}_i + 1) = 2kt}} \sum_{i=1}^{t} (\bar{y}_i - y_i)\mathbf{x}_i^T \mathbf{w} - \sum_{i=1}^{t} y_i \bar{y}_i. \tag{3}$$

**Partial AUC.** This measures the area under the ROC curve with the false positive rate restricted to the range $[0, \beta]$. This is in contrast to AUC that considers the entire range $[0, 1]$ of false positive rates. pAUC is useful in medical applications such as cancer detection where a small false positive rate is desirable. Let us extend notation to use the indicator $\mathbb{T}_{\beta,t}^-(\mathbf{x}, \mathbf{w})$ to select the top $\beta$ fraction of the *negatively* labeled points i.e. $\mathbb{T}_{\beta,t}^-(\mathbf{x}, \mathbf{w}) = 1$ iff $\left|\{j : y_j < 0, \mathbf{w}^\top \mathbf{x} > \mathbf{w}^\top \mathbf{x}_j\}\right| \leq \lceil \beta t_- \rceil$ where $t_-$ is the number of negatives. Then we define

$$\text{pAUC}(\mathbf{w}) = \sum_{i:y_i > 0} \sum_{j:y_j < 0} \mathbb{T}_{\beta,t}^-(\mathbf{x}_j, \mathbf{w}) \cdot \mathbb{I}[\mathbf{x}_i^\top \mathbf{w} \geq \mathbf{x}_j^\top \mathbf{w}]. \tag{4}$$

Let $\phi : \mathbb{R} \to \mathbb{R}_+$ be any convex, monotone, Lipschitz, classification surrogate. Then we can obtain convex surrogates for $\text{pAUC}(\mathbf{w})$ by replacing the indicator functions above with $\phi(\cdot)$.

$$\ell_{\text{pAUC}}(\mathbf{w}) = \sum_{i:y_i > 0} \sum_{j:y_j < 0} \mathbb{T}_{\beta,t}^-(\mathbf{x}_j, \mathbf{w}) \cdot \phi(\mathbf{x}_i^\top \mathbf{w} - \mathbf{x}_j^\top \mathbf{w}), \tag{5}$$

It can be shown [7, Theorem 4] that the structural surrogate for pAUC is equivalent to (5) with $\phi(c) = \max(0, 1 - c)$, the hinge loss function. In the next section we will develop an online learning framework for non-decomposable performance measures and instantiate loss functions such as $\ell_{\text{Prec}_{@k}}$ and $\ell_{\text{pAUC}}$ in our framework. Then in Section 4, we will develop stochastic gradient methods for non-decomposable loss functions and prove error bounds for the same. There we will focus on a much larger family of loss functions including $\text{Prec}_{@k}$, pAUC and F-measure.

# 3 Online Learning with Non-decomposable Loss Functions

We now present our online learning framework for non-decomposable loss functions. Traditional online learning takes place in several rounds, in each of which the player proposes some $\mathbf{w}_t \in \mathcal{W}$ while the adversary responds with a penalty function $\mathcal{L}_t : \mathcal{W} \to \mathbb{R}$ and a loss $\mathcal{L}_t(\mathbf{w}_t)$ is incurred. The goal is to minimize the *regret* i.e. $\sum_{t=1}^{T} \mathcal{L}_t(\mathbf{w}_t) - \arg\min_{\mathbf{w} \in \mathcal{W}} \sum_{t=1}^{T} \mathcal{L}_t(\mathbf{w})$. For point loss functions, the *instantaneous* penalty $\mathcal{L}_t(\cdot)$ is encoded using a data point $(\mathbf{x}_t, y_t) \in \mathbb{R}^d \times \{-1, 1\}$ as $\mathcal{L}_t(\mathbf{w}) = \ell_{\mathcal{P}}(\mathbf{x}_t, y_t, \mathbf{w})$. However, for (surrogates of) non-decomposable loss functions such as $\ell_{\text{pAUC}}$ and $\ell_{\text{Prec}@k}$ the definition of instantaneous penalty itself is not clear and remains a challenge.

To guide us in this process we turn to some properties of standard online learning frameworks. For point losses, we note that the best solution in hindsight is also the batch optimal solution. This is equivalent to the condition $\arg\min_{\mathbf{w} \in \mathcal{W}} \sum_{t=1}^{T} \mathcal{L}_t(\mathbf{w}) = \arg\min_{\mathbf{w} \in \mathcal{W}} \ell_{\mathcal{P}}(\mathbf{x}_{1:T}, y_{1:T}, \mathbf{w})$ for non-decomposable losses. Also, since the batch optimal solution is agnostic to the ordering of points, we should expect $\sum_{t=1}^{T} \mathcal{L}_t(\mathbf{w})$ to be invariant to permutations within the stream. By pruning away several naive definitions of $\mathcal{L}_t$ using these requirements, we arrive at the following definition:

$$\mathcal{L}_t(\mathbf{w}) = \ell_{\mathcal{P}}(\mathbf{x}_{1:t}, y_{1:t}, \mathbf{w}) - \ell_{\mathcal{P}}(\mathbf{x}_{1:(t-1)}, y_{1:(t-1)}, \mathbf{w}). \tag{6}$$

It turns out that the above is a very natural penalty function as it measures the amount of "extra" penalty incurred due to the inclusion of $\mathbf{x}_t$ into the set of points. It can be readily verified that $\sum_{t=1}^{T} \mathcal{L}_t(\mathbf{w}) = \ell_{\mathcal{P}}(\mathbf{x}_{1:T}, y_{1:T}, \mathbf{w})$ as required. Also, this penalty function seamlessly generalizes online learning frameworks since for point losses, we have $\ell_{\mathcal{P}}(\mathbf{x}_{1:t}, y_{1:t}, \mathbf{w}) = \sum_{i=1}^{t} \ell_{\mathcal{P}}(\mathbf{x}_i, y_i, \mathbf{w})$ and thus $\mathcal{L}_t(\mathbf{w}) = \ell_{\mathcal{P}}(\mathbf{x}_t, y_t, \mathbf{w})$. We note that our framework also recovers the model for online AUC maximization used in [13] and [14]. The notion of regret corresponding to this penalty is

$$R(T) = \frac{1}{T} \sum_{t=1}^{T} \mathcal{L}_t(\mathbf{w}_t) - \arg\min_{\mathbf{w} \in \mathcal{W}} \frac{1}{T} \ell_{\mathcal{P}}(\mathbf{x}_{1:T}, y_{1:T}, \mathbf{w}).$$

We note that $\mathcal{L}_t$, being the difference of two loss functions, is non-convex in general and thus, standard online convex programming regret bounds cannot be applied in our framework. Interestingly, as we show below, by exploiting structural properties of our penalty function, we can still get efficient low-regret learning algorithms, as well as online-to-batch conversion bounds in our framework.

## 3.1 Low Regret Online Learning

We propose an efficient Follow-the-Regularized-Leader (FTRL) style algorithm in our framework. Let $\mathbf{w}_1 = \arg\min_{\mathbf{w} \in \mathcal{W}} \|\mathbf{w}\|_2^2$ and consider the following update:

$$\mathbf{w}_{t+1} = \arg\min_{\mathbf{w} \in \mathcal{W}} \sum_{t=1}^{t} \mathcal{L}_t(\mathbf{w}) + \frac{\eta}{2} \|\mathbf{w}\|_2^2 = \arg\min_{\mathbf{w} \in \mathcal{W}} \ell_{\mathcal{P}}(\mathbf{x}_{1:t}, y_{1:t}, \mathbf{w}) + \frac{\eta}{2} \|\mathbf{w}\|_2^2 \qquad \text{(FTRL)}$$

We would like to stress that despite the non-convexity of $\mathcal{L}_t$, the FTRL objective is strongly convex if $\ell_{\mathcal{P}}$ is convex and thus the update can be implemented efficiently by solving a regularized batch problem on $\mathbf{x}_{1:t}$. We now present our regret bound analysis for the FTRL update given above.

**Theorem 1.** *Let $\ell_{\mathcal{P}}(\cdot, \mathbf{w})$ be a convex loss function and $\mathcal{W} \subseteq \mathbb{R}^d$ be a convex set. Assume w.l.o.g. $\|\mathbf{x}_t\|_2 \leq 1, \forall t$. Also, for the penalty function $\mathcal{L}_t$ in (6), let $|\mathcal{L}_t(\mathbf{w}) - \mathcal{L}_t(\mathbf{w}')| \leq G_t \cdot \|\mathbf{w} - \mathbf{w}'\|_2$, for all $t$ and all $\mathbf{w}, \mathbf{w}' \in \mathcal{W}$, for some $G_t > 0$. Suppose we use the update step given in ((FTRL)) to obtain $\mathbf{w}_{t+1}, 0 \leq t \leq T - 1$. Then for all $\mathbf{w}^*$, we have*

$$\frac{1}{T} \sum_{t=1}^{T} \mathcal{L}_t(\mathbf{w}_t) \leq \frac{1}{T} \ell_{\mathcal{P}}(\mathbf{x}_{1:T}, y_{1:T}, \mathbf{w}^*) + \|\mathbf{w}^*\|_2 \frac{\sqrt{2 \sum_{t=1}^{T} G_t^2}}{T}.$$

See Appendix A for a proof. The above result requires the penalty function $\mathcal{L}_t$ to be Lipschitz continuous i.e. be "stable" w.r.t. $\mathbf{w}$. Establishing this for point losses such as hinge loss is relatively straightforward. However, the same becomes non-trivial for non-decomposable loss functions as

$\mathcal{L}_t$ is now the difference of two loss functions, both of which involve $\Omega(t)$ data points. A naive argument would thus, only be able to show $G_t \leq O(t)$ which would yield vacuous regret bounds.

Instead, we now show that for the surrogate loss functions for Prec$_{@k}$ and pAUC, this Lipschitz continuity property does indeed hold. Our proofs crucially use a structural lemma given below that shows that sorted lists of inner products are Lipschitz at each fixed position.

**Lemma 2** (Structural Lemma). *Let* $\mathbf{x}_1, \ldots, \mathbf{x}_t$ *be* $t$ *points with* $\|\mathbf{x}_i\|_2 \leq 1 \, \forall t$. *Let* $\mathbf{w}, \mathbf{w}' \in \mathcal{W}$ *be any two vectors. Let* $z_i = \langle \mathbf{w}, \mathbf{x}_i \rangle - c_i$ *and* $z_i' = \langle \mathbf{w}', \mathbf{x}_i \rangle - c_i$, *where* $c_i \in \mathbb{R}$ *are constants independent of* $\mathbf{w}, \mathbf{w}'$. *Also, let* $\{i_1, \ldots, i_t\}$ *and* $\{j_1, \ldots, j_t\}$ *be ordering of indices such that* $z_{i_1} \geq z_{i_2} \geq \cdots \geq z_{i_t}$ *and* $z_{j_1}' \geq z_{j_2}' \geq \cdots \geq z_{j_t}'$. *Then for any 1-Lipschitz increasing function* $g : \mathbb{R} \to \mathbb{R}$ *(i.e.* $|g(u) - g(v)| \leq |u - v|$ *and* $u \leq v \Leftrightarrow g(u) \leq g(v)$), *we have,* $\forall k \, |g(z_{i_k}) - g(z_{j_k}')| \leq 3\|\mathbf{w} - \mathbf{w}'\|_2$.

See Appendix B for a proof. Using this lemma we can show that the Lipschitz constant for $\ell_{\text{Prec}@k}$ is bounded by $G_t \leq 8$ which gives us a $\mathcal{O}\left(\sqrt{\frac{1}{T}}\right)$ regret bound for Prec$_{@k}$ (see Appendix C for the proof). In Appendix D, we show that the same technique can be used to prove a stability result for the structural SVM surrogate of the Precision-Recall Break Even Point (PRBEP) performance measure [3] as well. The case of pAUC is handled similarly. However, since pAUC discriminates between positives and negatives, our previous analysis cannot be applied directly. Nevertheless, we can obtain the following regret bound for pAUC (a proof will appear in the full version of the paper).

**Theorem 3.** *Let* $T_+$ *and* $T_-$ *resp. be the number of positive and negative points in the stream and let* $\mathbf{w}_{t+1}$, $0 \leq t \leq T - 1$ *be obtained using the FTRL algorithm* ((FTRL)). *Then we have*

$$\frac{1}{\beta T_+ T_-} \sum_{t=1}^{T} \mathcal{L}_t(\mathbf{w}_t) \leq \min_{\mathbf{w} \in \mathcal{W}} \frac{1}{\beta T_+ T_-} \ell_{pAUC}(\mathbf{x}_{1:T}, y_{1:T}, \mathbf{w}) + \mathcal{O}\left(\sqrt{\frac{1}{T_+} + \frac{1}{T_-}}\right).$$

Notice that the above regret bound depends on both $T_+$ and $T_-$ and the regret becomes large even if one of them is small. This is actually quite intuitive because if, say $T_+ = 1$ and $T_- = T - 1$, an adversary may wish to provide the lone positive point in the last round. Naturally the algorithm, having only seen negatives till now, would not be able to perform well and would incur a large error.

## 3.2 Online-to-batch Conversion

To present our bounds we generalize our framework slightly: we now consider the stream of $T$ points to be composed of $T/s$ batches $\mathbf{Z}_1, \ldots, \mathbf{Z}_{T/s}$ of size $s$ each. Thus, the instantaneous penalty is now defined as $\mathcal{L}_t(\mathbf{w}) = \ell_{\mathcal{P}}(\mathbf{Z}_1, \ldots, \mathbf{Z}_t, \mathbf{w}) - \ell_{\mathcal{P}}(\mathbf{Z}_1, \ldots, \mathbf{Z}_{t-1}, \mathbf{w})$ for $t = 1 \ldots T/s$ and the regret becomes $R(T, s) = \frac{1}{T} \sum_{t=1}^{T/s} \mathcal{L}_t(\mathbf{w}_t) - \arg\min_{\mathbf{w} \in \mathcal{W}} \frac{1}{T} \ell_{\mathcal{P}}(\mathbf{x}_{1:T}, y_{1:T}, \mathbf{w})$. Let $\mathcal{R}_{\mathcal{P}}$ denote the population risk for the (normalized) performance measure $\mathcal{P}$. Then we have:

**Theorem 4.** *Suppose the sequence of points* $(\mathbf{x}_t, y_t)$ *is generated i.i.d. and let* $\mathbf{w}_1, \mathbf{w}_2, \ldots, \mathbf{w}_{T/s}$ *be an ensemble of models generated by an online learning algorithm upon receiving these* $T/s$ *batches. Suppose the online learning algorithm has a guaranteed regret bound* $R(T, s)$. *Then for* $\overline{\mathbf{w}} = \frac{1}{T/s} \sum_{t=1}^{T/s} \mathbf{w}_t$, *any* $\mathbf{w}^* \in \mathcal{W}$, $\epsilon \in (0, 0.5]$ *and* $\delta > 0$, *with probability at least* $1 - \delta$,

$$\mathcal{R}_{\mathcal{P}}(\overline{\mathbf{w}}) \leq (1 + \epsilon)\mathcal{R}_{\mathcal{P}}(\mathbf{w}^*) + R(T, s) + e^{-\Omega(s\epsilon^2)} + \tilde{\mathcal{O}}\left(\sqrt{\frac{s \ln(1/\delta)}{T}}\right).$$

*In particular, setting* $s = \tilde{\mathcal{O}}(\sqrt{T})$ *and* $\epsilon = \sqrt[4]{1/T}$ *gives us, with probability at least* $1 - \delta$,

$$\mathcal{R}_{\mathcal{P}}(\overline{\mathbf{w}}) \leq \mathcal{R}_{\mathcal{P}}(\mathbf{w}^*) + R(T, \sqrt{T}) + \tilde{\mathcal{O}}\left(\sqrt[4]{\frac{\ln(1/\delta)}{T}}\right).$$

We conclude by noting that for Prec$_{@k}$ and pAUC, $R(T, \sqrt{T}) \leq \mathcal{O}\left(\sqrt[4]{1/T}\right)$ (see Appendix E).

## 4  Stochastic Gradient Methods for Non-decomposable Losses

The online learning algorithms discussed in the previous section present attractive guarantees in the sequential prediction model but are required to solve batch problems at each stage. This rapidly

| **Algorithm 1 1PMB**: Single-Pass with Mini-batches | **Algorithm 2 2PMB**: Two-Passes with Mini-batches |
|---|---|
| **Input:** Step length scale $\eta$, Buffer $B$ of size $s$ | **Input:** Step length scale $\eta$, Buffers $B_+$, $B_-$ of size $s$ |
| **Output:** A good predictor $\mathbf{w} \in \mathcal{W}$ | **Output:** A good predictor $\mathbf{w} \in \mathcal{W}$ |
| 1: $\mathbf{w}_0 \leftarrow \mathbf{0}, B \leftarrow \phi, e \leftarrow 0$ | **Pass 1:** $B_+ \leftarrow \phi$ |
| 2: **while** stream not exhausted **do** | 1: Collect random sample of pos. $\mathbf{x}_1^+, \ldots, \mathbf{x}_s^+$ in $B_+$ |
| 3:     Collect $s$ data points $(\mathbf{x}_1^e, y_1^e), \ldots, (\mathbf{x}_s^e, y_s^e)$ in buffer $B$ | **Pass 2:** $\mathbf{w}_0 \leftarrow \mathbf{0}, B_- \leftarrow \phi, e \leftarrow 0$ |
| 4:     Set step length $\eta_e \leftarrow \frac{\eta}{\sqrt{e}}$ | 2: **while** stream of negative points not exhausted **do** |
| 5:     $\mathbf{w}_{e+1} \leftarrow \Pi_{\mathcal{W}} \left[ \mathbf{w}_e + \eta_e \nabla_{\mathbf{w}} \ell_{\mathcal{P}}(\mathbf{x}_{1:s}^e, y_{1:s}^e, \mathbf{w}_e) \right]$ //$\Pi_{\mathcal{W}}$ projects onto the set $\mathcal{W}$ | 3:     Collect $s$ negative points $\mathbf{x}_1^{e-}, \ldots, \mathbf{x}_s^{e-}$ in $B_-$ |
| 6:     Flush buffer $B$ | 4:     Set step length $\eta_e \leftarrow \frac{\eta}{\sqrt{e}}$ |
| 7:     $e \leftarrow e + 1$     //start a new epoch | 5:     $\mathbf{w}_{e+1} \leftarrow \Pi_{\mathcal{W}} \left[ \mathbf{w}_e + \eta_e \nabla_{\mathbf{w}} \ell_{\mathcal{P}}(\mathbf{x}_{1:s}^{e-}, \mathbf{x}_{1:s}^+, \mathbf{w}_e) \right]$ |
| 8: **end while** | 6:     Flush buffer $B_-$ |
| 9: **return** $\overline{\mathbf{w}} = \frac{1}{e} \sum_{i=1}^e \mathbf{w}_i$ | 7:     $e \leftarrow e + 1$     //start a new epoch |
|  | 8: **end while** |
|  | 9: **return** $\overline{\mathbf{w}} = \frac{1}{e} \sum_{i=1}^e \mathbf{w}_i$ |

becomes infeasible for large scale data. To remedy this, we now present memory efficient stochastic gradient descent methods for batch learning with non-decomposable loss functions. The motivation for our approach comes from mini-batch methods used to make learning methods for *point* loss functions amenable to distributed computing environments [15, 16], we exploit these techniques to offer scalable algorithms for *non-decomposable* loss functions.

**Single-pass Method with Mini-batches.** The method assumes access to a limited memory buffer and takes a pass over the data stream. The stream is partitioned into *epochs*. In each epoch, the method accumulates points in the stream, uses them to form gradient estimates and takes descent steps. The buffer is flushed after each epoch. Algorithm 1 describes the **1PMB** method. Gradient computations can be done using Danskin's theorem (see Appendix H).

**Two-pass Method with Mini-batches.** The previous algorithm is unable to exploit relationships between data points across epochs which may help improve performance for loss functions such as pAUC. To remedy this, we observe that several real life learning scenarios exhibit mild to severe label imbalance (see Table 2 in Appendix H) which makes it possible to store all or a large fraction of points of the rare label. Our two pass method exploits this by utilizing two passes over the data: the first pass collects all (or a random subset of) points of the rare label using some stream sampling technique [13]. The second pass then goes over the stream, restricted to the non-rare label points, and performs gradient updates. See Algorithm 2 for details of the **2PMB** method.

## 4.1 Error Bounds

Given a set of $n$ labeled data points $(\mathbf{x}_i, y_i), i = 1 \ldots n$ and a performance measure $\mathcal{P}$, our goal is to approximate the empirical risk minimizer $\mathbf{w}^* = \arg\min_{\mathbf{w} \in \mathcal{W}} \ell_{\mathcal{P}}(\mathbf{x}_{1:n}, y_{1:n}, \mathbf{w})$ as closely as possible. In this section we shall show that our methods **1PMB** and **2PMB** provably converge to the empirical risk minimizer. We first introduce the notion of uniform convergence for a performance measure.

**Definition 5.** *We say that a loss function $\ell$ demonstrates uniform convergence with respect to a set of predictors $\mathcal{W}$ if for some $\alpha(s, \delta) = poly\left(\frac{1}{s}, \log\frac{1}{\delta}\right)$, when given a set of $s$ points $\bar{\mathbf{x}}_1, \ldots, \bar{\mathbf{x}}_s$ chosen randomly from an arbitrary set of $n$ points $\{(\mathbf{x}_1, y_1), \ldots, (\mathbf{x}_n, y_n)\}$ then w.p. at least $1 - \delta$, we have*

$$\sup_{\mathbf{w} \in \mathcal{W}} |\ell_{\mathcal{P}}(\mathbf{x}_{1:n}, y_{1:n}, \mathbf{w}) - \ell_{\mathcal{P}}(\bar{\mathbf{x}}_{1:s}, \bar{y}_{1:s}, \mathbf{w})| \leq \alpha(s, \delta).$$

Such uniform convergence results are fairly common for decomposable loss functions such as the squared loss, logistic loss etc. However, the same is not true for non-decomposable loss functions barring a few exceptions [17, 10]. To bridge this gap, below we show that a large family of surrogate loss functions for popular non decomposable performance measures does indeed exhibit uniform convergence. Our proofs require novel techniques and do not follow from traditional proof progressions. However, we first show how we can use these results to arrive at an error bound.

**Theorem 6.** *Suppose the loss function $\ell$ is convex and demonstrates $\alpha(s, \delta)$-uniform convergence. Also suppose we have an arbitrary set of $n$ points which are randomly ordered, then the predictor*

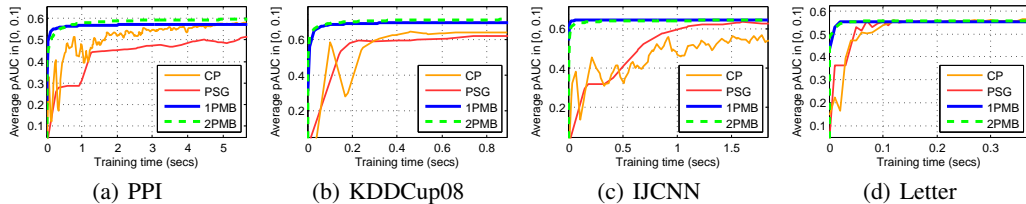

Figure 1: Comparison of stochastic gradient methods with the cutting plane (CP) and projected subgradient (PSG) methods on partial AUC maximization tasks. The epoch lengths/buffer sizes for **1PMB** and **2PMB** were set to 500.

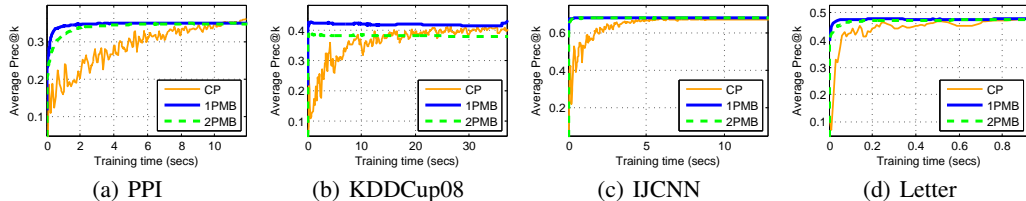

Figure 2: Comparison of stochastic gradient methods with the cutting plane (CP) method on Prec$_{@k}$ maximization tasks. The epoch lengths/buffer sizes for **1PMB** and **2PMB** were set to 500.

$\overline{\mathbf{w}}$ *returned by* **1PMB** *with buffer size $s$ satisfies w.p. $1 - \delta$,*

$$\ell_{\mathcal{P}}(\mathbf{x}_{1:n}, y_{1:n}, \overline{\mathbf{w}}) \leq \ell_{\mathcal{P}}(\mathbf{x}_{1:n}, y_{1:n}, \mathbf{w}_*) + 2\alpha\left(s, \frac{s\delta}{n}\right) + \mathcal{O}\left(\sqrt{\frac{s}{n}}\right)$$

We would like to stress that the above result does not assume i.i.d. data and works for arbitrary datasets so long as they are randomly ordered. We can show similar guarantees for the two pass method as well (see Appendix F). Using regularized formulations, we can also exploit logarithmic regret guarantees [18], offered by online gradient descent, to improve this result – however we do not explore those considerations here. Instead, we now look at specific instances of loss functions that possess the desired uniform convergence properties. As mentioned before, due to the combinatorial nature of these performance measures, our proofs do not follow from traditional methods.

**Theorem 7** (Partial Area under the ROC Curve). *For any convex, monotone, Lipschitz, classification surrogate $\phi : \mathbb{R} \to \mathbb{R}_+$, the surrogate loss function for the $(0, \beta)$-partial AUC performance measure defined as follows exhibits uniform convergence at the rate $\alpha(s, \delta) = \mathcal{O}\left(\sqrt{\log(1/\delta)/s}\right)$:*

$$\frac{1}{\lceil \beta n_- \rceil n_+} \sum_{i:y_i > 0} \sum_{j:y_j < 0} \mathbb{T}_{\beta,t}^-(\mathbf{x}_j, \mathbf{w}) \cdot \phi(\mathbf{x}_i^\top \mathbf{w} - \mathbf{x}_j^\top \mathbf{w})$$

See Appendix G for a proof sketch. This result covers a large family of surrogate loss functions such as hinge loss (5), logistic loss etc. Note that the insistence on including only top ranked negative points introduces a high degree of non-decomposability into the loss function. A similar result for the special case $\beta = 1$ is due to [17]. We extend the same to the more challenging case of $\beta < 1$.

**Theorem 8** (Structural SVM loss for Prec$_{@k}$). *The structural SVM surrogate for the Prec$_{@k}$ performance measure (see (3)) exhibits uniform convergence at the rate $\alpha(s, \delta) = \mathcal{O}\left(\sqrt{\log(1/\delta)/s}\right)$.*

We defer the proof to the full version of the paper. The above result can be extended to a large family of performances measures introduced in [3] that have been widely adopted [10, 19, 8] such as F-measure, G-mean, and PRBEP. The above indicates that our methods are expected to output models that closely approach the empirical risk minimizer for a wide variety of performance measures. In the next section we verify that this is indeed the case for several real life and benchmark datasets.

## 5 Experimental Results

We evaluate the proposed stochastic gradient methods on several real-world and benchmark datasets.

| Measure | 1PMB | 2PMB | CP |
|---|---|---|---|
| pAUC | 0.10 (68.2) | 0.15 (**69.6**) | 0.39 (62.5) |
| Prec$_{@k}$ | 0.49 (**42.7**) | 0.55 (38.7) | 23.25 (40.8) |

Table 1: Comparison of training time (secs) and accuracies (in brackets) of **1PMB**, **2PMB** and cutting plane methods for pAUC (in $[0, 0.1]$) and Prec$_{@k}$ maximization tasks on the KDD Cup 2008 dataset.

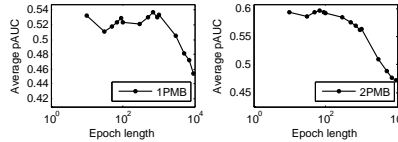

Figure 3: Performance of **1PMB** and **2PMB** on the PPI dataset with varying epoch/buffer sizes for pAUC tasks.

**Performance measures**: We consider three measures, 1) partial AUC in the false positive range $[0, 0.1]$, 2) Prec@$k$ with $k$ set to the proportion of positives (PRBEP), and 3) F-measure.

**Algorithms**: For partial AUC, we compare against the state-of-the-art cutting plane (CP) and projected subgradient methods (PSG) proposed in [7]; unlike the (online) stochastic methods considered in this work, the PSG method is a 'batch' algorithm which, at each iteration, computes a subgradient-based update over the entire training set. For Prec$_{@k}$ and F-measure, we compare our methods against cutting plane methods from [3]. We used structural SVM surrogates for all the measures.

**Datasets**: We used several data sets for our experiments (see Table 2 in Appendix H); of these, KDDCup08 is from the KDD Cup 2008 challenge and involves a breast cancer detection task [20], PPI contains data for a protein-protein interaction prediction task [21], and the remaining datasets are taken from the UCI repository [22].

**Parameters**: We used 70% of the data set for training and the remaining for testing, with the results averaged over 5 random train-test splits. Tunable parameters such as step length scale were chosen using a small validation set. The epoch lengths/buffer sizes were set to 500 in all experiments. Since a single iteration of the proposed stochastic methods is very fast in practice, we performed multiple passes over the training data (see Appendix H for details).

The results for pAUC and Prec$_{@k}$ maximization tasks are shown in the Figures 1 and 2. We found the proposed stochastic gradient methods to be several orders of magnitude faster than the baseline methods, all the while achieving comparable or better accuracies. For example, for the pAUC task on the KDD Cup 2008 dataset, the **1PMB** method achieved an accuracy of 64.81% within 0.03 seconds, while even after 0.39 seconds, the cutting plane method could only achieve an accuracy of 62.52% (see Table 1). As expected, the (online) stochastic gradient methods were faster than the 'batch' projected subgradient descent method for pAUC as well. We found similar trends on Prec$_{@k}$ (see Figure 2) and F-measure maximization tasks as well. For F-measure tasks, on the KDD Cup 2008 dataset, for example, the **1PMB** method achieved an accuracy of 35.92 within 12 seconds whereas, even after 150 seconds, the cutting plane method could only achieve an accuracy of 35.25.

The proposed stochastic methods were also found to be robust to changes in epoch lengths (buffer sizes) till such a point where excessively long epochs would cause the number of updates as well as accuracy to dip (see Figure 3). The **2PMB** method was found to offer higher accuracies for pAUC maximization on several datasets (see Table 1 and Figure 1), as well as be more robust to changes in buffer size (Figure 3). We defer results on more datasets and performance measures to the full version of the paper.

The cutting plane methods were generally found to exhibit a zig-zag behaviour in performance across iterates. This is because these methods solve the dual optimization problem for a given performance measure; hence the intermediate models do not necessarily yield good accuracies. On the other hand, (stochastic) gradient based methods directly offer progress in terms of the primal optimization problem, and hence provide good intermediate solutions as well. This can be advantageous in scenarios with a time budget in the training phase.

## Acknowledgements

The authors thank Shivani Agarwal for helpful comments. They also thank the anonymous reviewers for their suggestions. HN thanks support from a Google India PhD Fellowship.

## Footnotes

[1]An equivalent definition considers $k$ to be the *number* of top ranked points instead.

[2][3] uses a slightly modified, but equivalent, definition that considers labels to be Boolean.

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
