[Supplementary Material]

# Supplementary Material for the Paper entitled

## Online and Stochastic Gradient Methods for Non-decomposable Loss Functions

## A  Proof of Theorem 1

Broadly, we follow the proof structure of FTRL given in [1, 23]. We first observe that the "forward regret" analysis follows easily despite the non-convexity of $\mathcal{L}_t$. That is,

$$\sum_{t=1}^{T} \mathcal{L}_t(\mathbf{w}_{t+1}) \leq \mathbf{x}_{1:T}, y_{1:T}, \mathbf{w}_*) + \frac{\eta}{2}\|\mathbf{w}_*\|_2^2, \tag{7}$$

where $\mathbf{w}_* = \arg\min_{\mathbf{w}\in\mathcal{W}} \mathbf{x}_{1:T}, y_{1:T}, \mathbf{w})$. The proof of this statement can be found in [23, Theorem 7] and is reproduced below as Lemma 9 for completeness. Next, using strong convexity of the regularizer $\|\mathbf{w}\|_2^2$ and optimality of $\mathbf{w}_t$ and $\mathbf{w}_{t+1}$ for their respective update steps, we get:

$$\ell_{\mathcal{P}}(\mathbf{x}_{1:t}, y_{1:t}, \mathbf{w}_{t+1}) + \frac{\eta}{2}\|\mathbf{w}_{t+1} - \mathbf{w}_t\|_2^2 \quad \leq \quad \ell_{\mathcal{P}}(\mathbf{x}_{1:t}, y_{1:t}, \mathbf{w}_t)$$

$$\ell_{\mathcal{P}}(\mathbf{x}_{1:t-1}, y_{1:t-1}, \mathbf{w}_{t+1}) \quad \geq \quad \ell_{\mathcal{P}}(\mathbf{x}_{1:t-1}, y_{1:t-1}, \mathbf{w}_t) + \frac{\eta}{2}\|\mathbf{w}_{t+1} - \mathbf{w}_t\|_2^2,$$

which when subtracted, give us

$$\eta\|\mathbf{w}_{t+1} - \mathbf{w}_t\|_2^2 \leq \mathcal{L}_t(\mathbf{w}_t) - \mathcal{L}_t(\mathbf{w}_{t+1}) \leq G_t\|\mathbf{w}_{t+1} - \mathbf{w}_t\|_2, \tag{8}$$

where the last inequality follows using the Lipschitz continuity of $\mathcal{L}_t$. We now use the fact that

$$\sum_{t=1}^{T} \mathcal{L}_t(\mathbf{w}_t) = \sum_{t=1}^{T} \mathcal{L}_t(\mathbf{w}_{t+1}) + \sum_{t=1}^{T} (\mathcal{L}_t(\mathbf{w}_t) - \mathcal{L}_t(\mathbf{w}_{t+1})),$$

along with (7) and (8) to get

$$\sum_{t=1}^{T} \mathcal{L}_t(\mathbf{w}_t) \leq \mathbf{x}_{1:T}, y_{1:T}, \mathbf{w}_*) + \frac{\eta}{2}\|\mathbf{w}_*\|_2^2 + \frac{\sum_{t=1}^{T} G_t^2}{\eta}.$$

The result now follows by selecting $\eta = \sqrt{2\sum_{t=1}^{T} G_t^2 / \|\mathbf{w}_*\|_2^2}$.

**Lemma 9.** *For the setting described in Theorem 1, we have*

$$\sum_{t=1}^{T} \mathcal{L}_t(\mathbf{w}_{t+1}) \leq \mathbf{x}_{1:T}, y_{1:T}, \mathbf{w}_*) + \frac{\eta}{2}\|\mathbf{w}_*\|_2^2$$

*Proof.* Let $\mathcal{L}_0(\mathbf{w}) := \frac{\eta}{2}\|\mathbf{w}\|_2^2$. Thus, we can equivalently write the FTRL update in (FTRL) as

$$\mathbf{w}_{t+1} = \arg\min_{\mathbf{w}\in\mathcal{W}} \sum_{\tau=0}^{t} \mathcal{L}_\tau(\mathbf{w}).$$

Now, using the optimality of $\mathbf{w}_{t+1}$ at time $t$, we get

$$\sum_{\tau=0}^{t} \mathcal{L}_\tau(\mathbf{w}_{t+1}) \leq \sum_{\tau=0}^{t} \mathcal{L}_\tau(\mathbf{w}_*) \tag{9}$$

Combining this with the optimality of $\mathbf{w}_t$ at time $t - 1$, we get

$$\sum_{\tau=0}^{t-1} \mathcal{L}_\tau(\mathbf{w}_t) + \mathcal{L}_t(\mathbf{w}_{t+1}) \leq \sum_{\tau=0}^{t} \mathcal{L}_\tau(\mathbf{w}_{t+1}) \leq \sum_{\tau=0}^{t} \mathcal{L}_\tau(\mathbf{w}_*) \tag{10}$$

Repeating this argument gives us

$$\sum_{\tau=0}^{t} \mathcal{L}_\tau(\mathbf{w}_{\tau+1}) \leq \sum_{\tau=0}^{t} \mathcal{L}_\tau(\mathbf{w}_*),$$

which proves the result. $\qquad\square$

## B   Proof of Lemma 2

We consider the following four exhaustive cases in turn:

**Case 1.** $z_{i_k} \geq z_{j_k}$ and $z'_{j_k} \geq z'_{i_k}$
We have the following set of inequalities

$$\begin{aligned}
g(z_{i_k}) &= g(\langle \mathbf{w}, \mathbf{x}_{i_k} \rangle - c_i) \\
&\leq g(\langle \mathbf{w}', \mathbf{x}_{i_k} \rangle - c_i) + |\langle \mathbf{w} - \mathbf{w}', \mathbf{x}_{i_k} \rangle| \\
&\leq g(\langle \mathbf{w}', \mathbf{x}_{i_k} \rangle - c_i) + \|\mathbf{w} - \mathbf{w}'\|_2 \\
&= g(z'_{i_k}) + \|\mathbf{w} - \mathbf{w}'\|_2 \\
&\leq g(z'_{j_k}) + \|\mathbf{w} - \mathbf{w}'\|_2,
\end{aligned}$$

where the first inequality follows by the Lipschitz assumption, the second follows by Cauchy-Schwartz inequality and the last follows by the case assumption $z'_{j_k} \geq z'_{i_k}$ and the fact that $g$ is an increasing function. By renaming $i \leftrightarrow j$ and $\mathbf{w} \leftrightarrow \mathbf{w}'$, we also have $g(z'_{j_k}) \leq g(z_{i_k}) + \|\mathbf{w} - \mathbf{w}'\|_2$. This establishes the result for the specific case.

**Case 2.** $z_{i_k} \leq z_{j_k}$ and $z'_{j_k} \leq z'_{i_k}$
This case follows similar to the case above.

**Case 3.** $z_{i_k} \geq z_{j_k}$ and $z'_{j_k} \leq z'_{i_k}$
Using the above conditions $z_{j_k}$ does not belong to the top $k$ elements of $z_1, \ldots, z_t$, but both $z'_{i_k}$ and $z'_{j_k}$ belong to the top $k$ elements of $z'_1, \ldots, z'_t$. Using the pigeonhole principle, there exists an index $s$ such that $z_s \geq z_{i_k}$ but $z_s \leq z'_{j_k}$. Hence, using arguments similar to Case 1, we get the following two bounds:

$$|g(z'_{i_k}) - g(z_s)| \leq \|\mathbf{w} - \mathbf{w}'\|_2,$$
$$|g(z_s) - g(z'_{j_k})| \leq \|\mathbf{w} - \mathbf{w}'\|_2.$$

We also have $\left| g(z'_{i_k}) - g(z_{i_k}) \right| \leq |\langle \mathbf{w} - \mathbf{w}', \mathbf{x}_{i_k} \rangle| \leq \|\mathbf{w} - \mathbf{w}'\|_2$. Adding these three inequalities gives us the desired result.

**Case 4.** $z_{i_k} \leq z_{j_k}$ and $z'_{j_k} \geq z'_{i_k}$
This case follows similar to the case above.

These cases are exhaustive and we thus conclude the proof.

## C   Stability result for Prec$_{@k}$

**Lemma 10.** *Let $\ell_{Prec_{@k}}$ be the surrogate for Prec$_{@k}$ as defined in (3), $\|\mathbf{x}_t\|_2 \leq 1, \forall t$ and $\mathcal{L}_t$ be defined as in (6). Then $\forall \mathbf{w}, \mathbf{w}' \in \mathcal{W}, |\mathcal{L}_t(\mathbf{w}) - \mathcal{L}_t(\mathbf{w}')| \leq 8\|\mathbf{w} - \mathbf{w}'\|_2$.*

*Proof.* Recall that, the loss function corresponding to Prec$_{@k}$ is defined as:

$$\ell_{\text{Prec}_{@k}}(\mathbf{x}_{1:t}, y_{1:t}, \mathbf{w}) = \max_{\substack{q \in \{-1,1\}^t \\ \sum_i (q_i+1) = 2\lceil kt \rceil}} \sum_{i=1}^{t} (q_i - y_i)\mathbf{x}_i^T \mathbf{w} - \sum_{i=1}^{t} q_i y_i \tag{11}$$

$$= \max_{\substack{q \in \{-1,1\}^t \\ \sum_i (q_i+1) = 2\lceil kt \rceil}} \underbrace{\sum_{i=1}^{t} q_i \mathbf{x}_i^T \mathbf{w} - \sum_{i=1}^{t} q_i y_i}_{A(\mathbf{x}_{1:t}, y_{1:t}, \mathbf{w})} - \underbrace{\sum_{i=1}^{t} y_i \mathbf{x}_i^T \mathbf{w}}_{B(\mathbf{x}_{1:t}, y_{1:t}, \mathbf{w})} \qquad (12)$$

Since $B(\mathbf{x}_{1:t}, y_{1:t}, \mathbf{w})$ is a decomposable loss function, it can at most add a constant (because of the assumptions made by us, that constant can be shown to be no bigger than 1) to the Lipschitz constant of $\mathcal{L}_t$. Hence we concentrate on bounding the contribution of $A(\mathbf{x}_{1:t}, y_{1:t}, \mathbf{w})$ to the Lipschitz constant of $\mathcal{L}_t$. Define $z_i = \langle \mathbf{w}, \mathbf{x}_i \rangle - y_i$ and $z_i' = \langle \mathbf{w}', \mathbf{x}_i \rangle - y_i$. It will be useful to rewrite $A(\mathbf{x}_{1:t}, y_{1:t}, \mathbf{w})$ as follows (and drop mentioning the dependence on $\mathbf{x}_{1:t}$ for notational simplicity):

$$p_t(\mathbf{w}) = 2 \max_{\substack{q \in \{1,0\}^t \\ \sum_i q_i = \lceil kt \rceil}} \sum_{i=1}^{t} q_i z_i - \sum_{i=1}^{t} z_i. \qquad (13)$$

Similarly, we can define $p_{t-1}(\mathbf{w})$ as well. Now we have

$$\begin{aligned}
\mathcal{L}_t(\mathbf{w}) - \mathcal{L}_t(\mathbf{w}') &= p_t(\mathbf{w}) - p_{t-1}(\mathbf{w}) - p_t(\mathbf{w}') + p_{t-1}(\mathbf{w}') + y_t \mathbf{x}_t(\mathbf{w}' - \mathbf{w}) \\
&\le \underbrace{p_t(\mathbf{w}) - p_{t-1}(\mathbf{w}) - p_t(\mathbf{w}') + p_{t-1}(\mathbf{w}')}_{\Delta_t(\mathbf{w}, \mathbf{w}')} + \|\mathbf{w} - \mathbf{w}'\|_2
\end{aligned}$$

Our mail goal in the sequel will be to show that $\Delta_t(\mathbf{w}, \mathbf{w}') \le \mathcal{O}(\|\mathbf{w} - \mathbf{w}'\|_2)$ which shall establish the desired Lipschitz continuity result. Now for both vectors $\mathbf{w}, \mathbf{w}'$ and time instances $t-1, t$, let us denote the optimal assignments as follows:

$$a^t = \arg\max_{\substack{q \in \{1,0\}^t \\ \sum_i q_i = \lceil kt \rceil}} \sum_{i=1}^{t} q_i z_i, \qquad\qquad b^t = \arg\max_{\substack{q \in \{1,0\}^t \\ \sum_i q_i = \lceil kt \rceil}} \sum_{i=1}^{t} q_i z_i',$$

$$a^{t-1} = \arg\max_{\substack{q \in \{1,0\}^{(t-1)} \\ \sum_i q_i = \lceil k(t-1) \rceil}} \sum_{i=1}^{t-1} q_i z_i, \qquad\qquad b^{t-1} = \arg\max_{\substack{q \in \{1,0\}^{(t-1)} \\ \sum_i q_i = \lceil k(t-1) \rceil}} \sum_{i=1}^{t-1} q_i z_i'.$$

Also, define indices $1 \le i_r \le t-1$ and $1 \le j_s \le t-1$ as:

$$z_{i_1} \ge z_{i_2} \cdots \ge z_{i_{t-1}},$$
$$z_{j_1}' \ge z_{j_2}' \cdots \ge z_{j_{t-1}}'.$$

Now, note that (13) involves maximization of a linear function, hence the optimizing assignment $q$ will always lie on the boundary of the Boolean hypercube with the cardinality constraint. Hence, $a^t$ can be obtained by setting $a_{i_r}^t = 1$, $\forall 1 \le r \le \lceil kt \rceil$ and $a_{i_r}^t = 0$, $\forall r > \lceil kt \rceil$, similarly for $b_t$. We consider the following two cases and within each, four subcases which establish the result.

In the rest of the proof, all invocations of Lemma 2 shall use the identity function for $g(\cdot)$ and $c_i = y_i$. Clearly this satisfies the prerequisites of Lemma 2 since the identity function is 1-Lipschitz and increasing.

**Case 1** $\lceil kt \rceil = \lceil k(t-1) \rceil = \alpha$
Within this, we have the following four exhaustive subcases:

**Case 1.1** $z_t \le z_{i_\alpha}$ and $z_t' \le z_{j_\alpha}'$
The above condition implies that both $a_t^t = 0$ and $b_t^t = 0$. Furthermore, $a_{1:(t-1)}^t = a^{t-1}$ and $b_{1:(t-1)}^t = b^{t-1}$. As a result we have

$$\Delta_t(\mathbf{w}, \mathbf{w}') = -z_t + z_t' = -\langle \mathbf{w}, \mathbf{x}_t \rangle + \langle \mathbf{w}', \mathbf{x}_t \rangle \le \|\mathbf{w} - \mathbf{w}'\|_2.$$

**Case 1.2** $z_t > z_{i_\alpha}$ and $z_t' \le z_{j_\alpha}'$
The above condition implies that $a_t^t = 1$ and $b_t^t = 0$. Hence, $b_{1:(t-1)}^t = b^{t-1}$. Also, as $a_t^t$ is turned on, the cardinality constraint dictates that one previously positive index should be turned off. That is, $a_{i_\alpha}^t = 0$, but $a_{i_\alpha}^{t-1} = 1$. Finally, $a_{i_r}^t = a_{i_r}^{t-1}, r \ne$

$\alpha$ and $r < t$. Using the above observations, we have the following sequence of inequalities:

$$\begin{aligned}
\Delta_t(\mathbf{w}, \mathbf{w}') &= (2(z_t - z_{i_\alpha}) - z_t) - (0 - z_t') \\
&= (z_t - z_{i_\alpha}) + (z_t' - z_{i_\alpha}) \\
&= (z_t - z_t') + 2(z_t' - z_{i_\alpha}) \\
&\leq (z_t - z_t') + 2(z_{j_\alpha}' - z_{i_\alpha}) \\
&\leq 7 \|\mathbf{w} - \mathbf{w}'\|_2,
\end{aligned}$$

where the third inequality follows from the case assumptions and the final inequality follows from an application of Cauchy Schwartz inequality and Lemma 2.

**Case 1.3** $z_t \leq z_{i_\alpha}$ and $z_t' > z_{j_\alpha}'$

In this case, we can analyze similarly to get

$$\begin{aligned}
\Delta_t(\mathbf{w}, \mathbf{w}') &= (0 - z_t) - (2(z_t' - z_{j_\alpha}') - z_t') \\
&= (z_{j_\alpha}' - z_t) + (z_{j_\alpha}' - z_t') \\
&= (z_t' - z_t) + 2(z_{j_\alpha}' - z_t') \\
&\leq (z_t' - z_t) \\
&\leq 3 \|\mathbf{w} - \mathbf{w}'\|_2.
\end{aligned}$$

**Case 1.4** $z_t > z_{i_\alpha}$ and $z_t' > z_{j_\alpha}'$

In this case, both $a_t^t = 1$ and $b_t^t = 1$. Hence, both $a_{i_\alpha}^t = 0$ and $b_{j_\alpha}^t = 0$. The remaining terms of $a^t$ and $a^{t-1}$ (similarly for $b^t$ and $b^{t-1}$) remain the same. That is, we have

$$\begin{aligned}
\Delta_t(\mathbf{w}, \mathbf{w}') &= (2(z_t - z_{i_\alpha}) - z_t) - (2(z_t' - z_{j_\alpha}') - z_t') \\
&= (z_t - z_t') - 2(z_{i_\alpha} - z_{j_\alpha}') \\
&\leq 7 \|\mathbf{w} - \mathbf{w}'\|_2.
\end{aligned}$$

**Case 2** $\lceil kt \rceil = \lceil k(t-1) \rceil + 1 = \alpha$

Here again, we consider the following four exhaustive subcases:

**Case 2.1** $z_t \leq z_{i_\alpha}$ and $z_t' \leq z_{j_\alpha}'$

The above condition implies that $a_t^t = 0$ and $b_t^t = 0$. Also, one new positive is included in both $a^t$ and $b^t$, i.e., $a_{i_\alpha}^t = 1$ and $b_{j_\alpha}^t = 1$. The remaining entries of $a^t$ and $b^t$ remains the same. Hence,

$$\Delta_t(\mathbf{w}, \mathbf{w}') = (2z_{i_\alpha} - z_t) - (2z_{j_\alpha}' - z_t') = 2(z_{i_\alpha} - z_{j_\alpha}') - (z_t - z_t') \leq 9 \|\mathbf{w} - \mathbf{w}'\|_2.$$

**Case 2.2** $z_t > z_{i_\alpha}$ and $z_t' \leq z_{j_\alpha}'$

The above condition implies that $a_t^t = 1$ and $b_t^t = 0$. Also, $b_{j_\alpha}^t = 1$. The remaining entries of $a^t$ and $b^t$ remains the same. Hence we have

$$\begin{aligned}
\Delta_t(\mathbf{w}, \mathbf{w}') &= (2z_t - z_t) - (2z_{j_\alpha}' - z_t') \\
&= (z_t - z_{j_\alpha}') + (z_t' - z_{j_\alpha}') \\
&= (z_t - z_t') + 2(z_t' - z_{j_\alpha}') \\
&\leq 3 \|\mathbf{w} - \mathbf{w}'\|_2.
\end{aligned}$$

**Case 2.3** $z_t \leq z_{i_\alpha}$ and $z_t' > z_{j_\alpha}'$

In this case we have

$$\begin{aligned}
\Delta_t(\mathbf{w}, \mathbf{w}') &= (2z_{i_\alpha} - z_t) - (2z_t' - z_t') \\
&= (z_{i_\alpha} - z_t) + (z_{i_\alpha} - z_t') \\
&= (z_t' - z_t) + 2(z_{i_\alpha} - z_t') \\
&\leq (z_t' - z_t) + 2(z_{i_\alpha} - z_{j_\alpha}') \\
&\leq 7 \|\mathbf{w} - \mathbf{w}'\|_2.
\end{aligned}$$

**Case 2.4** $z_t > z_{i_\alpha}$ and $z'_t > z'_{j_\alpha}$

The above condition implies that $a^t_t = 1$ and $b^t_t = 1$. The remaining entries of $a^t$ and $b^t$ remains the same. Hence,

$$\Delta_t(\mathbf{w}, \mathbf{w}') = (2z_t - z_t) - (2z'_t - z'_t) = z_t - z'_t \leq 3 \left\| \mathbf{w} - \mathbf{w}' \right\|_2.$$

Taking the worst case Lipschitz constants from these 8 subcases and adding the contribution of $B(\mathbf{x}_{1:t}, y_{1:t}, \mathbf{w})$ concludes the proof. □

## D  Extension to Precision-Recall Break Even Point (PRBEP)

We note that the above discussion can easily be extended to prove stability results for the structural surrogate loss for the PRBEP performance measure [3]. Recall that the PRBEP measure essentially measures the precision (equivalently recall) of a predictor when thresholded at a point that equates the precision and recall. Since we have $\text{Prec} = \frac{\text{TP}}{\text{TP+FP}}$ and $\text{Rec} = \frac{\text{TP}}{\text{TP+FN}}$, the break even point is reached at a threshold where $\text{TP} + \text{FP} = \text{TP} + \text{FN}$. Notice that the left hand side equals the number of points that are predicted as positive whereas the right hand side equals the number of points that are actual positives.

Thus, the PRBEP is achieved at a threshold that predicts as many points as positive as there are actual positives which gives us the formal definition of this performance measure

$$\text{PRBEP}(\mathbf{w}) := \sum_{j : \mathbb{T}\left(\frac{t_+}{t}, t\right)(\mathbf{x}_j, \mathbf{w}) = 1} \mathbb{I}\left[y_j = 1\right]. \tag{14}$$

Note that this is equivalent to the definition of $\text{Prec}_{@k}$ with $k = \frac{t_+}{t}$. Correspondingly, we can also define the structural SVM surrogate for this performance measure as

$$\ell_{\text{PRBEP}}(\mathbf{w}) = \max_{\substack{\bar{\mathbf{y}} \in \{-1, +1\}^t \\ \sum_i (\bar{y}_i + 1) = 2t_+}} \sum_{i=1}^{t} (\bar{y}_i - y_i) \mathbf{x}_i^T \mathbf{w} - \sum_{i=1}^{t} y_i \bar{y}_i. \tag{15}$$

Given this, it is easy to see that the proof of Lemma 10 would apply to this case as well. The only difference in applying the analysis would be that Case 1 and its subcases would apply when $y_t < 0$ which is when the incoming point is negative and hence the number of actual positives in the stream does not go up. Case 2 and its subcases would apply when $y_t > 0$ in which case the number of points to be considered while calculating precision would have to be increased by 1.

## E  Online-to-batch Conversion

This section presents a proof of the regret bound in the batch model considered in Theorem 4 and a proof sketch of the online-to-batch conversion result. The full proof shall appear in the full version of the paper. We will consider in this section, the pAUC measure in the **2PMB** setting wherein positives are assumed to reside in the buffer and negatives are streaming in. The case of the $\text{Prec}_{@k}$ measure in the usual **1PMB** setting can be handled similarly. Additionally, we will show in Appendix G that for the case of pAUC, the contributions from a large enough buffer of randomly chosen positive points mimics the contributions of the entire population of positive points. Thus, for pAUC, it suffices to show the online-to-batch conversion bounds just with respect to the negatives. We clarify this further in the discussion.

### E.1  Regret Bounds in the Modified Framework

We prove the following lemma which will help us in instantiating our online-to-batch conversion proofs.

**Lemma 11.** *For the surrogate losses of $Prec_{@k}$ and pAUC, we have $R(T, s) \leq \sqrt{s} \cdot R(T)$*

*Proof.* The only thing we need to do is analyze one time step for changes in the Lipschitz constant. Fix a time step $t$ and let $\mathbf{Z}_t = \{\mathbf{x}_{t,1}, \mathbf{x}_{t,2}, \ldots, \mathbf{x}_{t,s}\}$. Also, let $g_t(\mathbf{w}, i) := \ell_{\mathcal{P}}(\mathbf{Z}_1, \ldots, \mathbf{Z}_{t-1}, \mathbf{x}_{t,1:i}, \mathbf{w})$ for any $i = 1 \ldots s$ (note that this gives us $g_t(\mathbf{w}, s) = \ell_{\mathcal{P}}(\mathbf{Z}_1, \ldots, \mathbf{Z}_t, \mathbf{w})$). Also let us abuse notation to denote $g_t(\mathbf{w}, 0) := \ell_{\mathcal{P}}(\mathbf{Z}_1, \ldots, \mathbf{Z}_{t-1}, \mathbf{w}) = g_{t-1}(\mathbf{w}, s)$. Let the Lipschitz constant in the model with batch size $s$ be denoted as $G_t^s$. Thus, we have $G_t^1 = G_t$, the Lipschitz constant for the problem in the original model (i.e. for $s = 1$). Then we have, for any $\mathbf{w}, \mathbf{w}' \in \mathcal{W}$,

$$
\begin{aligned}
|\mathcal{L}_t(\mathbf{w}) - \mathcal{L}_t(w')| &= |\ell_{\mathcal{P}}(\mathbf{Z}_{1:t}, \mathbf{w}) - \ell_{\mathcal{P}}(\mathbf{Z}_{1:t-1}, \mathbf{w}) - \ell_{\mathcal{P}}(\mathbf{Z}_{1:t}, \mathbf{w}') + \ell_{\mathcal{P}}(\mathbf{Z}_{1:t-1}, \mathbf{w}')| \\
&= |g_t(\mathbf{w}, s) - g_t(\mathbf{w}, 0) - g_t(\mathbf{w}', s) + g_t(\mathbf{w}', 0)| \\
&= \left| \sum_{i=1}^s g_t(\mathbf{w}, i) - g_t(\mathbf{w}, i-1) - g_t(\mathbf{w}', i) + g_t(\mathbf{w}', i-1) \right| \\
&\leq \sum_{i=1}^s |g_t(\mathbf{w}, i) - g_t(\mathbf{w}, i-1) - g_t(\mathbf{w}', i) + g_t(\mathbf{w}', i-1)| \\
&\leq \sum_{i=1}^s G_t \|\mathbf{w} - \mathbf{w}'\| = G_t \cdot s \|\mathbf{w} - \mathbf{w}'\|,
\end{aligned}
$$

where the first inequality follows by triangle inequality and the second inequality follows by a repeated application of the Lipschitz property of these loss functions in the original online model (i.e. with batch size $s = 1$). This establishes the Lipschitz constant in this model as $G_t^s \leq s \cdot G_t$. Now, the usual FTRL analysis gives us the following (note that there are only $T/s$ time steps now)

$$
\sum_{t=1}^{T/s} \mathcal{L}_t(\mathbf{w}_t) \leq \ell_{\mathcal{P}}(\mathbf{x}_{1:T}, y_{1:T}, \mathbf{w}_*) + \frac{\eta}{2} \|\mathbf{w}_*\|_2^2 + \frac{\sum_{t=1}^{T/s} (G_t^s)^2}{\eta} \leq \ell_{\mathcal{P}}(\mathbf{x}_{1:T}, y_{1:T}, \mathbf{w}_*) + 2s \|\mathbf{w}_*\|_2 \sqrt{\sum_{t=1}^{T/s} G_t^2},
$$

by setting $\eta$ appropriately. Now, for Prec$_{@k}$, $G_t \leq 8$. Thus, we have

$$
\frac{1}{T} \sum_{t=1}^{T/s} \mathcal{L}_t(\mathbf{w}_t) \leq \frac{1}{T} \ell_{\mathcal{P}}(\mathbf{x}_{1:T}, y_{1:T}, \mathbf{w}_*) + 6 \|\mathbf{w}_*\|_2 \sqrt{\frac{s}{T}},
$$

which establishes the result for Prec$_{@k}$. Similarly, for pAUC, we can show that the regret in the batch model does not worsen by more than a factor of $\sqrt{s}$. $\qquad \square$

### E.2 Online-to-batch Conversion for pAUC

We will consider the **2PMB** setting where negative points come as a stream and positive points reside in an in-memory buffer. At each trial $t$, the learner receives a batch of $s$ negative points $\mathbf{Z}_t^- = \{\mathbf{x}_{t,1}^-, \ldots, \mathbf{x}_{t,s}^-\}$ (we shall assume throughout, for simplicity, that $s\beta$ is an integer). Let us denote the loss w.r.t all the positive points in the buffer by $\phi_+ : \mathcal{W} \times \mathbb{R} \to [0, B]$. $\phi_+$ is defined using a loss function $g(\cdot)$ such as hinge loss or logistic loss as

$$
\phi_+(\mathbf{w}, c) = \frac{1}{B} \sum_{i=1}^B g(\mathbf{w}^\top \mathbf{x}_i^+ - c)
$$

For sake of brevity, we will abbreviate $\phi_+(\mathbf{w}, c)$ as $\phi_+(c)$, the reference to $\mathbf{w}$ being clear from context. We assume that $\phi_+$ is monotonically increasing (as is the case for hinge loss and logistic regression) and bounded i.e. for some fixed $B > 0$, we have, for all $\mathbf{w} \in \mathcal{W}, c \in \mathbb{R}, 0 \leq \phi_+(\mathbf{w}, c) \leq B$. The empirical (unnormalized) partial AUC loss for a model $\mathbf{w} \in \mathcal{W} \subseteq \mathbb{R}^d$ over the negative points received in $t$ trials is then given by

$$
\tilde{\ell}_{\text{pAUC}}(\mathbf{Z}_{1:t}^-, \mathbf{w}) = \sum_{\tau=1}^t \sum_{q=1}^s \mathbb{T}_{\beta,t}^-(\mathbf{x}_{\tau,q}^-, \mathbf{w}) \phi_+(\mathbf{w}^\top \mathbf{x}_{\tau,q}^-),
$$

where $\mathbb{T}_{\beta,t}^-(\mathbf{x}^-, \mathbf{w})$ is the (empirical) indicator function that is turned on whenever $\mathbf{x}^-$ appears in the top-$\beta$ fraction of all the negatives seen till now, ordered by $\mathbf{w}$, i.e. $\mathbb{T}_{\beta,t}^-(\mathbf{x}^-, \mathbf{w}) = 1$ whenever

$\left|\left\{\tau \in [t], q \in [s] : \mathbf{w}^\top \mathbf{x}^- > \mathbf{w}^\top \mathbf{x}^-_{\tau,q}\right\}\right| \leq ts\beta$. We similarly define a population version of this empirical loss function as

$$\widetilde{\mathcal{R}}_{\text{pAUC}}(\mathbf{w}) = \mathbb{E}_{\mathbf{x}^-}\left[\!\left[\mathbb{T}^-_\beta(\mathbf{x}^-, \mathbf{w})\,\phi_+(\mathbf{w}^\top \mathbf{x}^-)\right]\!\right],$$

where $\mathbb{T}^-_\beta(\mathbf{x}^-, \mathbf{w})$ is the population indicator function with $\mathbb{T}^-_\beta(\mathbf{x}^-, \mathbf{w}) = 1$ whenever $\mathbb{P}_{\widetilde{\mathbf{x}}^-}\left(\mathbf{w}^\top \widetilde{\mathbf{x}}^- > \mathbf{w}^\top \mathbf{x}^-\right) \leq \beta$. Also, we define $\mathcal{L}_t(\mathbf{w}) = \ell_{\text{pAUC}}(\mathbf{Z}^-_{1:t}, \mathbf{w}) - \ell_{\text{pAUC}}(\mathbf{Z}^-_{1:t-1}, \mathbf{w})$, with the regret of a learning algorithm that generates an ensemble of models $\mathbf{w}_1, \mathbf{w}_2, \ldots, \mathbf{w}_{T/s} \in \mathcal{W} \subseteq \mathbb{R}^d$ upon receiving $T/s$ batches of negative points $\mathbf{Z}^-_{1:T/s}$ defined as:

$$R(T, s) = \frac{1}{T}\sum_{t=1}^{T/s}\mathcal{L}_t(\mathbf{w}_t) - \arg\min_{\mathbf{w} \in \mathcal{W}}\frac{1}{T}\widetilde{\ell}_{\text{pAUC}}(\mathbf{Z}^-_{1:T/s}, \mathbf{w}).$$

Define $\beta_t = \mathbb{E}_{\mathbf{x}^-}\left[\!\left[\mathbb{T}^-_{\beta,t-1}(\mathbf{x}^-, \mathbf{w}_t)\right]\!\right]$ as the fraction of the population that can appear in the top $\beta$ fraction of the set of points seen till now, i.e. the fraction of the population for which the empirical indicator function is turned on, and

$$\mathcal{Q}_t(\mathbf{w}) = \mathbb{E}_{\mathbf{x}^-}\left[\!\left[\mathbb{T}^-_{\beta,t-1}(\mathbf{x}^-, \mathbf{w})\,\phi_+(\mathbf{w}^\top \mathbf{x}^-)\right]\!\right]$$

as the population partial AUC computed with respect to the empirical indicator function $\mathbb{T}^-_{\beta,t-1}$ (note that the population risk functional $\widetilde{\mathcal{R}}_{\text{pAUC}}(\mathbf{w})$ is computed with respect to $\mathbb{T}^-_\beta(\mathbf{x}^-, \mathbf{w})$, the population indicator function instead). We will also find it useful to define the following conditional expectation.

$$\widetilde{\mathcal{L}}_t(\mathbf{w}) = \mathbb{E}_{\mathbf{Z}^-_t}\left[\!\left[\mathcal{L}_t(\mathbf{w}) \,|\, \mathbf{Z}^-_{1:t-1}\right]\!\right].$$

We now present a proof sketch of the online-to-batch conversion result in Theorem 4 for pAUC.

**Theorem 12** (Online-to-batch Conversion for pAUC). *Suppose the sequence of negative points $\mathbf{x}^-_1, \ldots, \mathbf{x}^-_T$ is generated i.i.d.. Let us partition this sequence into $T/s$ batches of size $s$ and let $\mathbf{w}_1, \mathbf{w}_2, \ldots, \mathbf{w}_{T/s}$ be an ensemble of models generated by an online learning algorithm upon receiving these $T/s$ batches. Suppose the online learning algorithm has a guaranteed regret bound $R(T, s)$. Then for $\overline{\mathbf{w}} = \frac{1}{T/s}\sum_{t=1}^{T/s}\mathbf{w}_t$, any $\mathbf{w}^* \in \mathcal{W} \subseteq \mathbb{R}^d$, $\epsilon \in (0, 1]$ and $\delta > 0$, with probability at least $1 - \delta$,*

$$\widetilde{\mathcal{R}}_{pAUC}(\overline{\mathbf{w}}) \leq (1 + \epsilon)\widetilde{\mathcal{R}}_{pAUC}(\mathbf{w}^*) + \frac{1}{\beta}R(T, s) + e^{-\Omega(s\epsilon^2)} + \tilde{\mathcal{O}}\left(\sqrt{\frac{s\ln(1/\delta)}{T}}\right).$$

*In particular, setting $s = \tilde{\mathcal{O}}(\sqrt{T})$ and $\epsilon = \sqrt[4]{1/T}$ gives us, with probability at least $1 - \delta$,*

$$\widetilde{\mathcal{R}}_{pAUC}(\overline{\mathbf{w}}) \leq \widetilde{\mathcal{R}}_{pAUC}(\mathbf{w}^*) + \frac{1}{\beta}R(T, \sqrt{T}) + \tilde{\mathcal{O}}\left(\sqrt[4]{\frac{\ln(1/\delta)}{T}}\right).$$

*Proof (Sketch).* Fix $\epsilon \in (0, 0.5]$. We wish to bound the difference

$$(1 - \epsilon)s\beta\sum_{t=1}^{T/s}\widetilde{\mathcal{R}}_{\text{pAUC}}(\mathbf{w}_t) - T\beta\widetilde{\mathcal{R}}_{\text{pAUC}}(\mathbf{w}_*) \tag{16}$$

and do so by decomposing (16) into four terms as shown below.

$$(16) \leq \sum_{t=1}^{T/s}RE_t(\mathbf{w}_t) + MC(\mathbf{w}_{1:T/s}) + R(\mathbf{w}_{1:T/s}) + UC(\mathbf{w}_*),$$

where we have

$$UC(\mathbf{w}_*) = \widetilde{\ell}_{\text{pAUC}}(\mathbf{Z}^-_{1:T/s}, \mathbf{w}_*) - T\beta\widetilde{\mathcal{R}}_{\text{pAUC}}(\mathbf{w}_*) \qquad \text{(Uniform Convergence Term)}$$

$$R(\mathbf{w}_{1:T/s}) = \sum_{t=1}^{T/s}\mathcal{L}_t(\mathbf{w}_t) - \sum_{t=1}^{T/s}\mathcal{L}_t(\mathbf{w}_*) \qquad \text{(Regret Term)}$$

$$MC(\mathbf{w}_{1:T/s}) = \sum_{t=1}^{T/s} \widetilde{\mathcal{L}}_t(\mathbf{w}_t) - \sum_{t=1}^{T/s} \mathcal{L}_t(\mathbf{w}_t) \qquad \text{(Martingale Convergence Terms)}$$

$$RE_t(\mathbf{w}_t) = (1-\epsilon)s\beta\widetilde{\mathcal{R}}_{\text{pAUC}}(\mathbf{w}_t) - \widetilde{\mathcal{L}}_t(\mathbf{w}_t) \qquad \text{(Residual Error Terms)}$$

Note that the above has used the fact that $\tilde{\ell}_{\text{pAUC}}(\mathbf{Z}_{1:T/s}^-, \mathbf{w}_*) = \sum_{t=1}^{T/s} \mathcal{L}_t(\mathbf{w}_*)$. We will bound these terms in order below. First we look at the term $UC(\mathbf{w}_*)$. Bounding this simply requires a batch generalization bound of the form we prove in Theorem 7. Thus, we can show, that with probability $1 - \delta/3$, we have

$$UC(\mathbf{w}_*) \leq \mathcal{O}\left(\sqrt{T \log(1/\delta)}\right).$$

We now move on the term $R(\mathbf{w}_{1:T/s})$. This is simply bounded by the regret of the ensemble $\mathbf{w}_{1:T/s}$. This gives us

$$R(\mathbf{w}_{1:T/s}) \leq T \cdot R(T, s).$$

The next term we bound is $MC(\mathbf{w}_{1:T/s})$. Note that by definition of $\widetilde{\mathcal{L}}_t(\mathbf{w})$, if we define

$$v_t = \widetilde{\mathcal{L}}_t(\mathbf{w}_t) - \mathcal{L}_t(\mathbf{w}_t),$$

then the terms $\{v_t\}$ form a martingale difference sequence. Since $\left|\widetilde{\mathcal{L}}_t(\mathbf{w}_t) - \mathcal{L}_t(\mathbf{w}_t)\right| \leq \mathcal{O}(s)$, we get, by an application of the Azuma-Hoefding inequality, with probability at least $1 - \delta/3$,

$$MC(\mathbf{w}_{1:T/s}) \leq \mathcal{O}\left(s\sqrt{\frac{T}{s}\ln\frac{1}{\delta}}\right) = \mathcal{O}\left(\sqrt{sT\ln(1/\delta)}\right).$$

The last step requires us to bound the residual term $RE_t(\mathbf{w}_t)$ which will again require uniform convergence techniques. We shall show, that with probability, at least $1 - (\delta \cdot s/3T)$, we have

$$\beta_t \geq \beta - \tilde{\mathcal{O}}\left(\sqrt{\frac{\log\frac{1}{\delta}}{s(t-1)}}\right).$$

This shall allow us to show that with the same probability, we have

$$\mathcal{Q}_t(\mathbf{w}_t) - \widetilde{\mathcal{R}}_{\text{pAUC}}(\mathbf{w}_t) \leq \tilde{\mathcal{O}}\left(\sqrt{\frac{\log\frac{1}{\delta}}{s(t-1)}}\right).$$

The last ingredient in the proof shall involve showing that the following holds for any $\epsilon > 0$

$$\widetilde{\mathcal{L}}_t(\mathbf{w}_t) \geq (1-\epsilon)s\beta_t\mathcal{Q}_t(\mathbf{w}_t) - \Omega\left(s\exp(-s\beta_t^2\epsilon^2)\right)$$

Combining the above with a union bound will show us that, with probability at least $1 - \delta/3$,

$$\sum_{i=1}^{T/s} RE_t(\mathbf{w}_t) \leq \mathcal{O}\left(T\exp(-s\epsilon^2)\right) + \tilde{\mathcal{O}}\left(\sqrt{sT\log(1/\delta)}\right)$$

A final union bound and some manipulations would then establish the claimed result. □

## F  Proof of Theorem 6

The proof proceeds in two parts: the first part uses the fact that the **1PMB** method essentially simulates the GIGA method of [24] with the non-decomposable loss function and the second part uses the uniform convergence properties of the loss function to establish the error bound. To proceed, let us set up some notation. Consider the $e^{\text{th}}$ epoch of the **1PMB** algorithm. Let us denote the set of points considered in this epoch by $X_e = \{x_1^e, \ldots, x_s^e\}$. With this notation it is clear that the **1PMB** algorithm can be said to be performing online gradient descent with respect to the instantaneous loss functions $\mathcal{L}_e(\mathbf{w}) = \mathcal{L}(X_e, \mathbf{w}) := \ell_{\mathcal{P}}(\mathbf{x}_{1:s}^e, y_{1:s}^e, \mathbf{w})$.

Since the loss function $\mathcal{L}_e(\mathbf{w})$ is convex, the standard analysis for online convex optimization would apply under mild boundedness assumptions on the domain and the (sub)gradients of the loss function. Since there are $n/s$ epochs (assuming for simplicity that $n$ is a multiple of $s$), this allows us to use the standard regret bounds [24] to state the following:

$$\frac{s}{n} \sum_{e=1}^{n/s} \mathcal{L}_e(\mathbf{w}_e) \leq \frac{s}{n} \sum_{e=1}^{n/s} \mathcal{L}_e(\mathbf{w}_*) + \mathcal{O}\left(\sqrt{\frac{s}{n}}\right).$$

Now we will invoke uniform convergence properties of the loss function. However, doing so requires clarifying certain aspects of the problem setting. The statement of Theorem 6 assumes only a random ordering of training data points whereas uniform convergence properties typically require i.i.d. samples. We reconcile this by noticing that all our uniform convergence proofs use the Hoeffding's lemma to establish statistical convergence and that the Hoeffding's lemma holds when random variables are sampled without replacement as well (e.g. see [25]). Since a random ordering of the data provides, for each epoch, a uniformly random sample without replacement, we are able to invoke the uniform convergence proofs.

Thus, if we denote $\mathcal{L}(\mathbf{w}) := \ell_{\mathcal{P}}(\mathbf{x}_{1:n}, y_{1:n}, \mathbf{w})$, then by using the uniform convergence properties of the loss function, for every $e$, with probability at least $1 - \frac{s\delta}{n}$, we have $\mathcal{L}_e(\mathbf{w}_e) \geq \mathcal{L}(\mathbf{w}_e) - \alpha\left(s, \frac{s\delta}{n}\right)$ as well as $\mathcal{L}_e(\mathbf{w}_*) \leq \mathcal{L}(\mathbf{w}_*) + \alpha\left(s, \frac{s\delta}{n}\right)$. Applying the union bound and Jensen's inequality gives us, with probability at least $1 - \delta$, the desired result:

$$\mathcal{L}(\overline{\mathbf{w}}) \leq \frac{s}{n} \sum_{e=1}^{n/s} \mathcal{L}(\mathbf{w}_e) \leq \mathcal{L}(\mathbf{w}_*) + 2\alpha\left(s, \frac{s\delta}{n}\right) + \mathcal{O}\left(\sqrt{\frac{s}{n}}\right).$$

We note that we can use similar arguments as above to give error bounds for the **2PMB** procedure as well. Suppose $\bar{\mathbf{x}}_{1:s_+}^+$ and $\bar{\mathbf{x}}_{1:s_-}^-$ are the positive and negative points sampled in the process (note that here the number of positive and negatives points (i.e. $s_+$ and $s_-$ respectively) are random quantities as well). Also suppose $\mathbf{x}_{1:n_+}^+$ and $\mathbf{x}_{1:n_-}^-$ are the positive and negative points in the population. Then recall that Definition 5 requires, for a uniform (but possibly without replacement) sample,

$$\sup_{\mathbf{w} \in \mathcal{W}} \left| \ell_{\mathcal{P}}(\mathbf{x}_{1:n_+}^+, \mathbf{x}_{1:n_-}^-, \mathbf{w}) - \ell_{\mathcal{P}}(\bar{\mathbf{x}}_{1:s_+}^+, \bar{\mathbf{x}}_{1:s_-}^-, \mathbf{w}) \right| \leq \tilde{\mathcal{O}}\left(\alpha(s, \delta)\right).$$

To prove bounds for **2PMB**, we require that for arbitrary choice of $s_+, s_- \geq \Omega(s)$, when $\bar{\mathbf{x}}_{1:s_+}^+$ and $\bar{\mathbf{x}}_{1:s_-}^-$ are chosen separately and uniformly (but yet again possibly without replacement) from $\mathbf{x}_{1:n_+}^+$ and $\mathbf{x}_{1:n_-}^-$ respectively, we still obtain a similar result as above. Since the first pass and each epoch of the second pass provide such a sample, we can use this result to prove error bounds for the **2PMB** procedure. We defer the detailed arguments for such results to the full version of the paper.

We however note that the proof of Theorem 7 below does indeed prove such a result for the pAUC loss function by effectively proving (see Section G.1) the following two results

$$\sup_{\mathbf{w} \in \mathcal{W}} \left| \ell_{\mathcal{P}}(\mathbf{x}_{1:n_+}^+, \bar{\mathbf{x}}_{1:s_-}^-, \mathbf{w}) - \ell_{\mathcal{P}}(\bar{\mathbf{x}}_{1:s_+}^+, \bar{\mathbf{x}}_{1:s_-}^-, \mathbf{w}) \right| \leq \tilde{\mathcal{O}}\left(\alpha(s, \delta)\right)$$

$$\sup_{\mathbf{w} \in \mathcal{W}} \left| \ell_{\mathcal{P}}(\mathbf{x}_{1:n_+}^+, \mathbf{x}_{1:n_-}^-, \mathbf{w}) - \ell_{\mathcal{P}}(\mathbf{x}_{1:n_+}^+, \bar{\mathbf{x}}_{1:s_-}^-, \mathbf{w}) \right| \leq \tilde{\mathcal{O}}\left(\alpha(s, \delta)\right).$$

# G   Uniform Convergence Bounds for Partial Area under the ROC Curve

In this section we present a proof sketch of Theorem 7 which we restate below for convenience.

**Theorem 13.** *Consider any convex, monotonic and Lipschitz classification surrogate $\phi : \mathbb{R} \to \mathbb{R}_+$. Then the loss function for the $(0, \beta)$-partial AUC performance measure defined as follows exhibits uniform convergence at the rate $\alpha(s) = \tilde{\mathcal{O}}(1/\sqrt{s})$:*

$$\ell_{\mathcal{P}}(\mathbf{x}_{1:n}, y_{1:n}, \mathbf{w}) = \frac{1}{\beta n_+ n_-} \sum_{i=1}^{n} \mathbb{I}\left[y_i > 0\right] \sum_{j=1}^{n} \mathbb{I}\left[y_j < 0\right] \mathbb{T}_{\beta,n}^-(\mathbf{x}_j, \mathbf{w}) \phi\left(\mathbf{w}^\top (\mathbf{x}_i - \mathbf{x}_j)\right),$$

*where $n_+ = |\{i : y_i > 0\}|$ and $n_- = |\{i : y_i < 0\}|$.*

*Proof (Sketch).* We shall use the notation $\hat{\mathbb{T}}_{\beta,s}^-$ to denote the indicator function for the top $\beta$ fraction of the negative elements in the smaller sample of size $s$. Thus, over the smaller sample $(\bar{\mathbf{x}}_1, \bar{y}_1) \ldots (\bar{\mathbf{x}}_s, \bar{y}_s)$, the pAUC is calculated as

$$\ell_{\mathcal{P}}(\bar{\mathbf{x}}_{1:s}, \bar{y}_{1:s}, \mathbf{w}) = \frac{1}{\beta s_+ s_-} \sum_{i=1}^{s} \mathbb{I}\left[\bar{y}_i > 0\right] \sum_{j=1}^{s} \mathbb{I}\left[\bar{y}_j < 0\right] \hat{\mathbb{T}}_{\beta,s}^-(\bar{\mathbf{x}}_j, \mathbf{w}) \phi\left(\mathbf{w}^\top(\bar{\mathbf{x}}_i - \bar{\mathbf{x}}_j)\right).$$

Our goal would be to show that with probability at least $1 - \delta$, for all $\mathbf{w} \in \mathcal{W}$

$$|\ell_{\mathcal{P}}(\mathbf{x}_{1:n}, y_{1:n}, \mathbf{w}) - \ell_{\mathcal{P}}(\bar{\mathbf{x}}_{1:s}, \bar{y}_{1:s}, \mathbf{w})| \leq \tilde{\mathcal{O}}\left(\frac{1}{\sqrt{s}}\right)$$

We shall demonstrate this by establishing the following three statements:

1. For any fixed $\mathbf{w} \in \mathcal{W}$, w.h.p., we have $|\ell_{\mathcal{P}}(\mathbf{x}_{1:n}, y_{1:n}, \mathbf{w}) - \ell_{\mathcal{P}}(\bar{\mathbf{x}}_{1:s}, \bar{y}_{1:s}, \mathbf{w})| \leq \tilde{\mathcal{O}}\left(\frac{1}{\sqrt{s}}\right)$

2. For any two $\mathbf{w}, \mathbf{w}' \in \mathcal{W}$, we have $|\ell_{\mathcal{P}}(\mathbf{x}_{1:n}, y_{1:n}, \mathbf{w}) - \ell_{\mathcal{P}}(\mathbf{x}_{1:n}, y_{1:n}, \mathbf{w}')| \leq \mathcal{O}\left(\|\mathbf{w} - \mathbf{w}'\|_2\right)$

3. For any two $\mathbf{w}, \mathbf{w}' \in \mathcal{W}$, we have $|\ell_{\mathcal{P}}(\bar{\mathbf{x}}_{1:s}, \bar{y}_{1:s}, \mathbf{w}) - \ell_{\mathcal{P}}(\bar{\mathbf{x}}_{1:s}, \bar{y}_{1:s}, \mathbf{w}')| \leq \mathcal{O}\left(\|\mathbf{w} - \mathbf{w}'\|_2\right)$

With these three results established, we would be able to conclude the proof by an application of a standard covering number argument. We now prove these three statements in parts.

### G.1 Part 1: Pointwise Convergence for pAUC

Fix a predictor $\mathbf{w} \in \mathcal{W}$ and $S_+$ and $S_-$ denote the set of positive and negative samples. We shall assume that $s_+, s_- \geq \Omega(s)$ which holds with high probability. Denote, for any $\mathbf{x}_i$ such that $y_i > 0$,

$$\ell^+(\mathbf{x}_i, \mathbf{w}) = \frac{1}{\beta n_-} \sum_{j=1}^{n} \mathbb{I}\left[y_j < 0\right] \mathbb{T}_{\beta,n}^-(\mathbf{x}_j, \mathbf{w}) \phi\left(\mathbf{w}^\top(\mathbf{x}_i - \mathbf{x}_j)\right),$$

and for any $\bar{\mathbf{x}}_i \in S_+$,

$$\ell_{S_-}^+(\bar{\mathbf{x}}_i, \mathbf{w}) = \frac{1}{\beta s_-} \sum_{j=1}^{s} \mathbb{I}\left[\bar{y}_j < 0\right] \hat{\mathbb{T}}_{\beta,s}^-(\bar{\mathbf{x}}_j, \mathbf{w}) \phi\left(\mathbf{w}^\top(\bar{\mathbf{x}}_i - \bar{\mathbf{x}}_j)\right).$$

Notice that $\ell_{\mathcal{P}}(\bar{\mathbf{x}}_{1:s}, \bar{y}_{1:s}, \mathbf{w}) = \frac{1}{n_+} \sum_{i=1}^{n} \mathbb{I}\left[y_i > 0\right] \ell^+(\mathbf{x}_i, \mathbf{w})$ and $\ell_{\mathcal{P}}(\bar{\mathbf{x}}_{1:s}, \bar{y}_{1:s}, \mathbf{w}) = \frac{1}{s_+} \sum_{i=1}^{s} \mathbb{I}\left[\bar{y}_i > 0\right] \ell_{S_-}^+(\bar{\mathbf{x}}_i, \mathbf{w})$. We shall now show the following holds w.h.p. over $S_-$:

1. For any $\mathbf{x}_i$ such that $y_i > 0$, $\left|\ell^+(\mathbf{x}_i, \mathbf{w}) - \ell_{S_-}^+(\mathbf{x}_i, \mathbf{w})\right| \leq \tilde{\mathcal{O}}\left(\frac{1}{\sqrt{s}}\right)$.

2. $\frac{1}{n_+} \left|\sum_{i=1}^{n} \mathbb{I}\left[y_i > 0\right] \ell^+(\mathbf{x}_i, \mathbf{w}) - \mathbb{I}\left[y_i > 0\right] \ell_{S_-}^+(\mathbf{x}_i, \mathbf{w})\right| \leq \tilde{\mathcal{O}}\left(\frac{1}{\sqrt{s}}\right)$.

3. $\left|\frac{1}{n_+} \sum_{i=1}^{n} \mathbb{I}\left[y_i > 0\right] \ell_{S_-}^+(\mathbf{x}_i, \mathbf{w}) - \frac{1}{s_+} \sum_{i=1}^{s} \mathbb{I}\left[\bar{y}_i > 0\right] \ell_{S_-}^+(\bar{\mathbf{x}}_i, \mathbf{w})\right| \leq \tilde{\mathcal{O}}\left(\frac{1}{\sqrt{s}}\right)$.

The second part follows from the first part by an application of the triangle inequality. The third part also can be shown to hold by an application of Hoeffding's inequality and other arguments. This leaves the first part for which we provide a proof in the full version of the paper.

### G.2 Parts 2 and 3: Establishing an $\epsilon$-net for pAUC

For simplicity, we assume that the domain is finite. This does not affect the proof in any way since it still allows the domain to be approximated arbitrary closely by an $\epsilon$-net of (arbitrarily) large size.

| Dataset | Data Points | Features | Positives |
|---------|-------------|----------|-----------|
| KDDCup08 | 102,294 | 117 | 0.61% |
| PPI | 240,249 | 85 | 1.19% |
| Letter | 20,000 | 16 | 3.92% |
| IJCNN | 141,691 | 22 | 9.57% |

Table 2: Statistics of datasets used.

However, we note that we can establish the same result for infinite domains as well, but choose not to for sake of simplicity. We prove the second part, the proof of the first part being similar. We have

$$
|\ell_{\mathcal{P}}(\mathbf{x}_{1:n}, y_{1:n}, \mathbf{w}) - \ell_{\mathcal{P}}(\mathbf{x}_{1:n}, y_{1:n}, \mathbf{w}')| = \frac{1}{s_+} \left| \sum_{i=1}^{s} \mathbb{I}\left[\bar{y}_i > 0\right] \ell_{S_-}^{+}(\mathbf{x}_i, \mathbf{w}) - \mathbb{I}\left[\bar{y}_i > 0\right] \ell_{S_-}^{+}(\mathbf{x}_i, \mathbf{w}') \right|
$$

$$
\leq \frac{1}{s_+} \sum_{i=1}^{s} \left| \mathbb{I}\left[\bar{y}_i > 0\right] \left( \ell_{S_-}^{+}(\mathbf{x}_i, \mathbf{w}) - \ell_{S_-}^{+}(\mathbf{x}_i, \mathbf{w}') \right) \right|
$$

$$
\leq \mathcal{O}\left( \|\mathbf{w} - \mathbf{w}'\|_2 \right),
$$

using Lemma 2 with $g(a) = \phi(\mathbf{w}^\top \mathbf{x}_i - a)$ and $c_i = 0$. This concludes the proof. $\qquad \square$

# H   Methodology for implementing 1PMB and 2PMB for pAUC tasks

In this section we clarify the mechanisms used to implement the **1PMB** and **2PMB** routines. Going as per the dataset statistics (see Table 2), we will consider the variant of the **2PMB** routine with the positive class as the rare class. Recall the definition of the surrogate loss function for pAUC (5)

$$
\ell_{\text{pAUC}}(\mathbf{w}) = \sum_{i:y_i>0} \sum_{j:y_j<0} \mathbb{T}_{\beta,t}^{-}(\mathbf{x}_j, \mathbf{w}) \cdot h(\mathbf{x}_i^\top \mathbf{w} - \mathbf{x}_j^\top \mathbf{w}).
$$

We now rewrite this in a slightly different manner. Define, for any $i : y_i > 0$

$$
\ell_{S_-}^{+}(x_i, \mathbf{w}) = \sum_{j:y_j<0} \mathbb{T}_{\beta,t}^{-}(\mathbf{x}_j, \mathbf{w}) \cdot h(\mathbf{x}_i^\top \mathbf{w} - \mathbf{x}_j^\top \mathbf{w}),
$$

so that we can write $\ell_{\text{pAUC}}(\mathbf{w}) = \sum_{i:y_i>0} \ell_{S_-}^{+}(x_i, \mathbf{w})$. This shows that a subgradient to $\ell_{\text{pAUC}}(\mathbf{w})$ can be found by simply finding and summing up, subgradients for $\ell_{S_-}^{+}(x_i, \mathbf{w})$. For now, fix an $i$ such that $y_i > 0$ and define $g(a) = h(\mathbf{x}_i^\top \mathbf{w} - a)$. Using the properties of the hinge loss function, it is clear that $g(a)$ is an increasing function of $a$. Since $\ell_{S_-}^{+}(x_i, \mathbf{w})$ is defined on the top ranked $\lceil \beta t_- \rceil$ negatives, we can, using the monotonicity of $g(\cdot)$, equivalently write it as follows. Let $\mathcal{Z}_\beta = \binom{S_-}{\lceil \beta t_- \rceil}$ be the set of all sets of negative points of negative training points of size $\lceil \beta t_- \rceil$. Then we can write

$$
\ell_{S_-}^{+}(x_i, \mathbf{w}) = \max_{S \in \mathcal{Z}_\beta} \sum_{\mathbf{x}^- \in S} g(\mathbf{x}^{-\top} \mathbf{w})
$$

Since the maximum in the above formulation is achieved at $S = \left\{ j : y_j < 0, \mathbb{T}_{\beta,t}^{-}(\mathbf{x}_j, \mathbf{w}) = 1 \right\}$, by Danskin's theorem (see, for example [26]), we get the following result: let $\mathbf{v}_{ij} \in \delta h(\mathbf{x}_i^\top \mathbf{w} - \mathbf{x}_j^\top \mathbf{w})$ be a subgradient to the hinge loss function, then for the following vector

$$
\mathbf{v}_i := \sum_{j:y_j<0} \mathbb{T}_{\beta,t}^{-}(\mathbf{x}_j, \mathbf{w}) \cdot \mathbf{v}_{ij},
$$

we have $\mathbf{v}_i \in \delta\ell_{S_-}^{+}(x_i, \mathbf{w})$ and consequently, for $\mathbf{v} := \sum_{i:y_i>0} \mathbf{v}_i$, we have $\mathbf{v} \in \delta\ell_{\text{pAUC}}(\mathbf{w})$. This gives us a straightforward way to implement **1PMB**: for each epoch, we take all the negatives in that epoch, filter out the top $\beta$ fraction of them according to the scores assigned to them by the current iterate $\mathbf{w}_e$ and then calculate the (sub)gradients between all the positives in that epoch and these filtered negatives. This takes at most $\mathcal{O}\left( s \log s \right)$ time per epoch.