[Reviews · NeurIPS 2014]

Submitted by Assigned_Reviewer_12

The authors present a framework for online optimization of non-decomposable loss functions (those that do not decompose as a sum over data points). Their idea is to write L_t(w) = l(w, x_{1:t}) - l(w, x_{1:t-1}). While L_t(w) in general will not be convex, its cumulative sum will be, which turns out to be enough for the FTRL analysis to hold. However, we still need to establish stability properties for L_t(w), which is harder than for decomposable loss functions. The authors provide a lemma that establishes sufficient conditions for stability, and show how it applies in the case that the loss function is precision@k. They then turn their attention to providing a gradient-based variant of their algorithm for the typical case that FTRL is too slow. Under relatively general conditions they establish a convergence rate of N^(-1/4) of their method and provide empirical support that the method works well.

Quality: the results are for the most part solid and convincing. *However*, there is one large issue, which is that I do not see why linearizing the (now non-convex) losses should still lead to an upper bound on the original regret. I would like the authors to comment on this in their rebuttal as it is an important point that affects my overall evaluation.

Clarity: the paper is very well-written, with the exception of the description of the gradient-based optimization (Algorithms 1 and 2) where some notation is not defined and it is not clear how to compute the gradients in an efficient manner.

Originality: the paper is fairly original.

Significance: the paper is very significant; it opens the door to a line of future research of online learning in the non-decomposable setting.

Questions for authors:
1. Why can the gradients be computed efficiently for Algorithms 1 and 2?
2. The theory suggests s should be sqrt(N), but in the experiments it seems like the best s are closer to O(1). What is the intuition for this? Are the theoretical results loose? Do you expect the "right" convergence rate to be N^(-1/4) or N^(-1/2)?

Minor comment:
You wrote $T_+T_-$ where you meant to write $T_+ + T_-$ in the statement of Theorem 3.
Summary: Convincing and well-written paper that presents an extension of the online learning framework. The authors should explain why the gradient can be computed efficiently and does not affect the regret.

UPDATE: Thanks to the authors for their clear and concise feedback. It has addressed most of my concerns and as such I am raising my score from 7 to 8.

Submitted by Assigned_Reviewer_44

This paper considers non-decomposable loss functions (focusing on two specific loss functions) in various settings. The authors propose an online framework that attempts to incrementally learn with these loss functions. The authors also propose two stochastic gradient decent methods for these loss functions that operate in a mini-batch mode. The convergence of these SGD is analyzed, and their efficiency is demonstrated empirically.

The presentation of the paper was mostly clear, with a couple of caveats—some of the notation was introduced a bit abruptly (a bit more explanation could expedite the reader’s understanding), and almost all the proofs and mathematical justifications were deferred to other papers or the appendix. Recognizing that there are space constraints, it still could help to have some proof sketches or ideas and more description of the key mathematical steps, particularly those deferred to other papers (e.g. (5)).

The proposed online framework seemed to be of very little significance as it required a separate optimization problem to be solved at every time iteration using all data from all previous iterations. In this sense, it arguably wasn't even a true online algorithm. This setup raised questions as to the actual benefits this online algorithm provides over the traditional batch setting.

I found the more impactful part of the paper to be the proposed stochastic gradient descent methods using mini-batch processing with non-decomposable loss functions, and the promising empirical results showing the efficiency of these algorithms for several basic experiments. The proposed algorithms were relatively simple and not particularly novel in themselves, although the 2PMB algorithm did incorporate a new idea for addressing label imbalance. However the paper would have benefited from a more detailed discussion/analysis of 2PMB (e.g. the random sampling process, the situations where it would perform well or poorly). The more novel part of this section of the paper was the analysis showing the convergence rates of these algorithms to the empirical risk minimizer. However, the authors only performed this analysis for the two specific loss functions considered in this paper, Prec@k and pAUC, so in its current state this paper is likely to only be incrementally impactful.
Summary: This paper develops an online framework for learning with non-decomposable loss functions, however the significance of this contribution is greatly limited by the fact the online algorithm must store and process all previously-revealed data points at every time iteration. This paper also proposes two stochastic gradient descent algorithms for working with non-decomposable losses—the authors provide promising analysis and empirical results demonstrating the efficiency of these algorithms, however this is likely an incremental impact as only two specific loss functions were considered.

Submitted by Assigned_Reviewer_45

Summary:
This paper studies online learning techniques for optimizing non-decomposable loss functions, such as precision@k and partial AUC, aiming to enable incremental learning as well as design scalable solvers for batch learning tasks. They prove sublinear regret and online to batch conversion bounds for the proposed algorithms. This paper develops scalable stochastic gradient descent solvers for non-decomposable loss functions and validates its efficiency via empirical studies.

Quality:
This paper proposes a practical online learning framework for non-decomposable loss functions. It presents two algorithms with mini batches, and gives theoretical guarantee with sublinear regret and online to batch conversion bounds. Empirically, the two proposed algorithms are compared with two existing batch learning algorithms, and found better results on several real-world data sets. In general, the work appears to be very similar to the series of studies in [13,14]. However, the authors did not clearly clarify the differences and contributions as compared to the similar work in [13,14], making it difficult to justify the novel contributions.

Clarity:
The manuscript is generally well written and easy to follow, except for some minor typos. For example, In line 170, it should be “not clear and remains” instead of “not clear an remains”.

Originality:
This paper extends the ideas in [13, 14] to tackle the general online learning tasks with non-decomposable loss functions. Instead of optimizing AUC, the authors consider two types of other losses, precision@k and partial AUC, and propose two specific online learning algorithms (single-pass and two-pass online learning) using the idea of mini-batches (which are essentially very similar to the buffering techniques as used in [13,14]). The extensions are OK, but not particularly novel.

Significance:
This paper addresses an important problem, online learning with non-decomposable loss functions, which could be more difficult than regular online learning with pointwise loss functions. The paper has presented two new online/stochastic learning algorithms for optimizing partial AUC and precision@k, and found better results in comparison to two existing batch learning algorithms. The limitations of this work is that the authors did not justify the significance of the proposed algorithms over the existing baselines in literature (e.g., [13,14]).

Comments on specific issues:
1. The empirical study has not been done in a thorough way. Since the paper aims to address online learning tasks, it is not sufficient to compare only with two batch learning algorithms. In particular, one would expect a direct comparison of the proposed algorithms with some closely related online learning algorithms in literature (e.g., [13,14]).
2. The two proposed algorithms (1PMB and 2PMB) have their obvious drawbacks although they seem to outperform the batch learning algorithms. In particular, for the first algorithm could fail to exploit relationships between data points, as admitted by the authors; for the second algorithm, simply collect a (single) random sample of positive example B+ may be neither effective (if sampling size s is too small) nor memory efficient (if sampling size s is too large). I think the buffer sampling treatments proposed in [13, 14] seem to be more practical to address such drawbacks of the proposed algorithms in this work.
3. Tuning parameters is not well explained. The authors only stated “all tunable parameters were chosen using a small validation set”. As the proposed algorithm depends on several important parameters such as the buffer size s of B+, B-, the step scale length \eta, which could affect the performance. However, there are no discussions or experiments for examining how these parameters.
4. The empirical results fail to validate efficacy of the proposed two-pass algorithm (2PMB). It seems that 2PBM is similar or even worse than the Single-pass algorithm (1PMB) for most cases. Since this algorithm needs to take multiple passes, the value of the 2PMB algorithm seems in doubt.
Summary: This work presents an online (stochastic) gradient descent framework for non-decomposable loss functions, and shows sublinear regret and online to batch conversion bounds for the proposed online learning algorithms. The proposed algorithms are not particularly novel as the similar idea has been explored in literature (e.g., [13,14]), although the bounds may be somewhat new and the empirical results seem positive (though empirical study is not thorough).
Author Feedback
Author rebuttal: We thank all reviewers for the detailed comments. Below are responses to the main comments.

Focus of paper: We would like to stress that the paper focusses on non-decomposable loss functions that are not handled by methods designed for point and pair-wise losses such as AUC [13,14]. Indeed, the buffer sampling methods used in [13,14] do not apply directly to the more complex loss in our paper. For non-decomposable losses, even defining regret is not self-evident. We design a generic framework for online learning with non-decomposable losses, provide tractable FTRL style algorithms with sublinear regret in this setting and also prove online-batch conversion bounds. Next, we use our intuition from the online algorithms to provide mini-batch style methods for stochastic settings with provable convergence rates for a WIDE RANGE of losses. Our methods perform significantly better than state-of-the-art methods.

Reviewer 12:

Linearization: In the online setting (Sec 3), we do not linearize the loss function and instead, directly use the FTRL analysis which, coupled with our "stability" argument, yields sublinear regret bounds. In the stochastic setting (Sec 4) where data arrives in an i.i.d. fashion, we are able to work directly with the batch loss function which is convex (see $\ell_P$ line 308). Here, we obtain accurate gradient estimates using mini-batches, and provide error bounds using uniform convergence arguments (Thm 7,8). We also stress that the uniform convergence bounds are novel and do not follow from traditional analyses.

Computation of gradient for stochastic methods:
Since the loss functions (Eq. 1) can be written as a maximum of convex functions, the gradients for these losses can be computed efficiently by applying Danskin's theorem. Appendix M provides details for pAUC loss. The gradient computation can be done in O(s log(s)) time for pAUC and Prec@K.

Epoch lengths (s): Our methods do indeed seem robust to epoch lengths which indicates that our bounds can be further tightened, possibly under certain low-noise assumptions.

T_+T_- in Thm 3: In the pAUC loss function, the number of effective terms is O(T_+T_-) (see Eq. 5); hence the normalization term in Thm 3. With Prec@K, the number of effective terms is O(T), hence a different normalization term.

Reviewer 44:

'Only two loss functions considered for stochastic methods':
We stress that this is not correct. While the online FTRL algorithms proposed were analyzed for two popular losses, we consider *SEVERAL LOSSES* for the stochastic methods, including pAUC, Prec@K, PRBEP, and structSVM based losses for a variety of other performance measures that are functions of TPR and TNR, such as F-measure and G-mean. Please see Thm 8 and the paragraph that follows; details are provided in Appendix J,K,L.

In subsequent experiments, we found that even for F-measure, our stochastic method handily outperforms the baseline cutting plane method [8]. On KDD08, 1PMB achieved (F1: 35.92, 12 sec) whereas CP could only get (F1: 35.25, >150 sec). We will report these in the paper.

Significance of our online framework/FTRL:
Online learning is primarily concerned with low regret. For non-decomposable losses, tractable algorithms with sublinear regret were not known earlier. We make progress in this direction by giving a novel regret definition and providing tractable (poly-time in T,d) algorithms with sublinear regret. Indeed, the literature has several FTRL-style algorithms as well as online algorithms that go so far as to solve intractable/NP-hard problems in pursuit of low regret (Dani et. al. Stochastic Linear Opt ... COLT 08) (Rakhlin et. al. Online Learning: Random Averages ... NIPS 10).

Reviewer 45:

1. Difference from online AUC optimization [13,14]:
We stress that our work is *NOT AN EXTENSION* of [13,14] which consider losses that do decompose into pairs. We consider non-decomposable losses such as pAUC and Prec@K which, in several applications (e.g. ranking problems), are preferred over decomposable losses e.g. AUC. Non-decomposability brings in several other technical issues which disallow the use of standard techniques prevalent for AUC maximization and other decomposable losses. E.g. the buffer sampling methods in [13,14] are specific to AUC since they make crucial use of the *decomposable pair-wise* structure of AUC, and cannot be directly used here. This is why technical elements such as those presented in Thm 6,7,8 had to be developed from scratch.

It is also well known [2,3,8] that training with decomposable losses (e.g. AUC [13,14]) does not lead to good performance w.r.t pAUC/Prec@K. We thus focus on and compare against state-of-the-art algorithms that optimize losses such as pAUC/Prec@K. We also believe that the uniform convergence bounds derived for the losses considered here are indeed novel, and do not follow from similar bounds for AUC [14].

We will certainly make this clear in the main text.

2. 'Empirical study not thorough': We disagree with the comment that we do not have a study of parameters. We indeed study the influence of epoch length on the two proposed stochastic methods, where we show that our methods are fairly robust to different epoch sizes; see Fig 3 and the last para in Sec 5. We can certainly include a more detailed discussion on parameters.

3. Details of parameter tuning: The epoch length was set to 500 in all experiments; this is mentioned in the captions of Fig 1 & 2. For the 2PMB algorithms, the positive buffer size was set to 500. Only the step-size \eta was tuned (from the range {2^-3, ... 2^3}) using a validation set of size 5000.

4. 1PMB vs 2PMB: 2PMB does perform better or similar to 1PMB on pAUC tasks (see Fig 1; Fig 4 in App N). For Prec@K, 2PMB is sometimes slower than 1PMB; we attribute this to the specific structure of Prec@K which does not compare positive and negatives directly, and hence does not benefit from having a separate positive buffer.